# Using a Panchromatic Image to Improve Hyperspectral Unmixing

**Simon Rebeyrol [1,2,*]** , **Yannick Deville [2]** , **Véronique Achard [1]** **and Xavier Briottet [1]** **and Stephane May [3]**

[1] ONERA "The French Aerospace Lab", Département Optique et Techniques Associées (DOTA), 2 Av. Edouard Belin, 31055 Toulouse, France; veronique.achard@onera.fr (V.A.); xavier.briottet@onera.fr (X.B.)

[2] IRAP, Université de Toulouse, UPS-CNRS-CNES, 14 Av. Edouard Belin, 31400 Toulouse, France; yannick.deville@irap.omp.eu

[3] Centre National D'études Spatiales (CNES), 18 Av. Edouard Belin, CEDEX 9, 31401 Toulouse, France; stephane.may@cnes.fr

[*] Correspondence: simon.rebeyrol@onera.fr

**Abstract:** Hyperspectral unmixing is a widely studied field of research aiming at estimating the pure material signatures and their abundance fractions from hyperspectral images. Most spectral unmixing methods are based on prior knowledge and assumptions that induce limitations, such as the existence of at least one pure pixel for each material. This work presents a new approach aiming to overcome some of these limitations by introducing a co-registered panchromatic image in the unmixing process. Our method, called Heterogeneity-Based Endmember Extraction coupled with Local Constrained Non-negative Matrix Factorization (HBEE-LCNMF), has several steps: a first set of endmembers is estimated based on a heterogeneity criterion applied on the panchromatic image followed by a spectral clustering. Then, in order to complete this first endmember set, a local approach using a constrained non-negative matrix factorization strategy, is proposed. The performance of our method, in regards of several criteria, is compared to those of state-of-the-art methods obtained on synthetic and satellite data describing urban and periurban scenes, and considering the French HYPXIM/HYPEX2 mission characteristics. The synthetic images are built with real spectral reflectances and do not contain a pure pixel for each endmember. The satellite images are simulated from airborne acquisition with the spatial and spectral features of the mission. Our method demonstrates the benefit of a panchromatic image to reduce some well-known limitations in unmixing hyperspectral data. On synthetic data, our method reduces the spectral angle between the endmembers and the real material spectra by 46% compared to the Vertex Component Analysis (VCA) and N-finder (N-FINDR) methods. On real data, HBEE-LCNMF and other methods yield equivalent performance, but, the proposed method shows more robustness over the data sets compared to the tested state-of-the-art methods. Moreover, HBEE-LCNMF does not require one to know the number of endmembers.

**Keywords:** hyperspectral; unmixing; panchromatic; satellite; HYPXIM; HYPEX2; heterogeneity; Endmember Extraction; Local Constrained Non-negative Matrix Factorization; NMF; LCNMF

## 1. Introduction

For several decades, Earth remote sensing has demonstrated its potential to monitor our environment. In particular, the hyperspectral (HS) imaging technique is able to sample the spectral radiance over the Visible, Near InfraRed (VNIR) and Short-Wave InfraRed (SWIR) spectral domains to retrieve the spectral reflectance of the surface of a given material. The spectral behavior is thus analyzed

to determine the status or the chemical composition of each element of the scene. For most of remote sensing hyperspectral cameras onboard a space platform, like the future HYPXIM/HYPEX-2 [1,2] with an 8 m Ground Sampling Distance (GSD) or PRISMA [3], HYPERION [4], or EnMAP [5] with a 30 m GSD, or onboard an aircraft, like Apex [6] or Hyspex [7], or AVIRIS-NG [8] with a GSD of a few meters depending on the acquisition altitude, the spatial resolution is not sufficient to ensure the presence of a single material in each pixel. Thus, the area covered by a pixel is most of the time composed of several materials, making the correct material identification difficult, as shown in Figure 1.

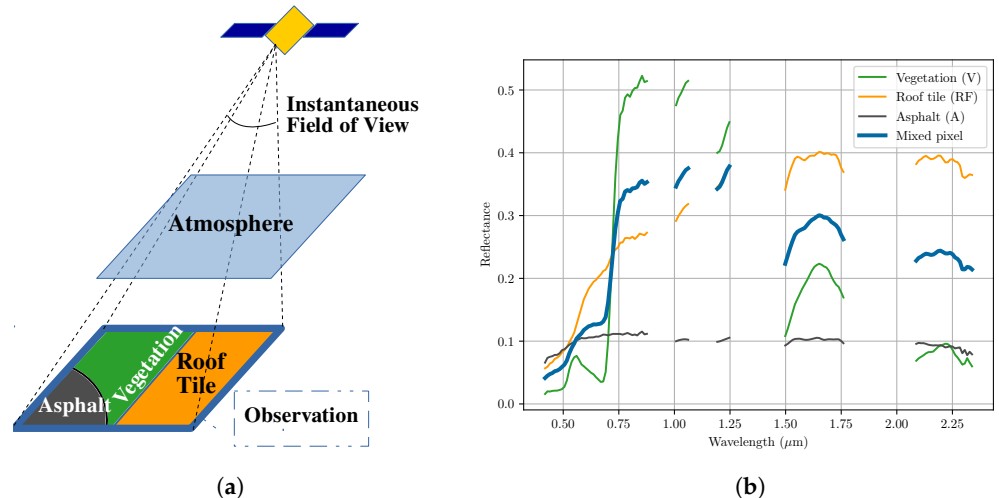

(**a**)    (**b**)

**Figure 1.** Example of a mixed pixel: (**a**) sketch of a mixed pixel projected at ground level and composed of three materials: asphalt, vegetation, and tile; (**b**) spectral signatures of each material and the equivalent spectral reflectance of the pixel.

To make the material identification possible, a blind source separation strategy [9] is applied to retrieve the pure material spectral signatures, i.e., the endmembers, and their relative abundance fractions in each pixel of a hyperspectral image. This problem has been widely studied for decades and is called hyperspectral unmixing in the remote sensing community. Many hyperspectral unmixing methods already exist [10] and are classified in a variety of families depending on the mixing model, the assumptions made and the used framework. Mostly, unmixing methods assume the well-known linear mixing model [11] since it fairly explains most of the macroscopic mixing phenomena for flat surfaces [12]. However, non-linear mixing models have also been developed to consider multiple scattering [13,14] at a macroscopic level. A few mixing models have also been proposed to address the spectral variability of the materials present in the image. This phenomenon can be due to a change of illumination [15] or the slope of the surface, the varying chemical composition of the materials or their surface conditions [16,17]. This paper will focus on the standard linear mixing model (LMM) and, as a consequence, will not consider these non-linearity or spectral variability effects. The LMM describes each observed spectrum from a pixel $n$ as the sum of the endmembers weighted by their abundance fractions as in Equation (1):

$$\vec{y_n} = \sum_{p=1}^{P} x_{np} \vec{s_p} + \vec{w_n} \qquad x_{np}, \vec{s_p} \geq 0 \tag{1}$$

with

$$\sum_{p=1}^{P} x_{np} = 1, \tag{2}$$

where $P$ is the number of endmembers in the scene, $x_{np}$ is the abundance fraction for pixel $n$ and material $p$, $\vec{s_p}$ is a column vector corresponding to the pure material signature number $p$ sampled

over $N_\lambda$ spectral bands, $\overrightarrow{w_n}$ represents the noise from pixel $n$ sampled over $N_\lambda$ spectral bands, and $\overrightarrow{y_n}$ is a column vector containing the observed spectrum from pixel $n$ sampled over $N_\lambda$ spectral bands. With the flat landscape assumption, each abundance fraction is defined as the ratio of the area covered by the pure material in the pixel to the area covered by the pixel, leading to the Abundance Sum-to-one Constraint (ASC, see Equation (2)). This constraint has led to a large family of unmixing methods based on geometrical analysis. Thus, the observed spectra are contained within a $P-1$ dimensional simplex in which their vertices are the endmembers. This family of unmixing methods is divided in two classes. The methods of the first class aim at finding all the vertices in the observed pixels assuming the presence of at least one pure pixel, i.e., a pixel containing the spectral contribution of only one pure material, for each pure material. Popular pure pixel-based methods are the Vertex Component Analysis (VCA) [18], N-FINDR [19], and the Automatic Target Generation Process (ATGP) [20]. However, the pure pixel assumption is often invalid in regards of the GSD commonly used in hyperspectral acquisitions because pure pixels are available only for a few pure materials. A second class of geometrical methods, which does not need the presence of pure pixels, aims at estimating the endmembers by fitting a minimum volume simplex enclosing the data with such methods as the Minimum-Volume Enclosing Simplex (MVES) [21], the Minimum Volume Simplex Analysis (MVSA) [22], and the Simplex Identification via Split Augmented Lagrangian (SISAL) [23]. In order to correctly fit the simplex, the data must contain observed spectra on each facet of the simplex. Like the pure pixel assumption, this hypothesis does not hold on real data and these methods are highly sensitive to their initialization. The Bayesian framework has also been studied to develop statistical hyperspectral unmixing approaches, such as the Joint Bayesian Endmember Extraction and Linear Unmixing method [24], referred to as JBEELU hereafter. Those methods require the knowledge of probability densities to model the endmembers and abundance fractions and yield a heavy computational cost. Other geometrically-based and statistical methods can be found in Reference [10]. Another popular family of unmixing methods relies on the Non-negative Matrix Factorization (NMF) framework [25]. Hyperspectral data are well suited to the NMF since the endmembers and abundance fractions are non-negative. But, the NMF is a non-convex optimization problem and can lead to a vast amount of solutions, most of which are unacceptable. To address this issue, one can apply additional prior knowledge or assumptions to the mixing model by constraining the optimization criterion [26–34]. Nevertheless, these NMF methods are highly sensitive to their initialization, especially when applied to an entire image subject to spectral variability [16,35,36]. New approaches have also been proposed, like in the K-Means Sparse Coding Dictionary (KMSCD) algorithm [37], which relies on a K-means clustering [38] to initialize a Sparse Coding Dictionary method [39]. These approaches exhibit the benefit of a clustering step in hyperspectral unmixing.

In addition to these unmixing methods, local unmixing strategies have also been considered to reduce the influence of the intra-class spectral variability phenomenon. In this regard, Canham et al. [40] proposed an unmixing method using sliding windows to divide a HS image into multiple regions to estimate the endmembers in each region. The method then performs a clustering to assign the unmixed local spectra to a global class. More recently, Drumetz et al. [41] developed an algorithm which uses a segmentation technique to select the local regions and performs a collaborative sparse regression to linearly unmix the whole image.

Finally, the above unmixing methods require the number of endmembers to be known beforehand. Since this number is rarely known, methods have been proposed to estimate it, such as HYSIME [42] and HFC [43]. More methods can be found in Reference [44–47]. Although these methods offer a relatively accurate estimation of the number of endmembers, a change in the number of materials can change the endmembers extracted by most methods and has a direct impact on the abundance estimates.

Since all the previously described methods show limitations due to the assumptions made and the data features, new hyperspectral unmixing strategies combining synchronized and distinct acquisitions with different characteristics may be relevant. For example, Uezato et al. [48] used LIght Detection And Ranging (LIDAR) data to unmix hyperspectral images, and Karoui [33], Benkouider et al. [49]

developed two algorithms taking in account multispectral data with a smaller GSD. Multispectral/hyperspectral and panchromatic/multispectral data fusion may also use hyperspectral unmixing techniques in their process [33,50–54]. Moreover, multiple satellite missions combining two co-registered hyperspectral and panchromatic (PAN) cameras, the latter with a better GSD than the former, are studied or even operational. For example, the PRISMA mission [3], is shipping two push-broom cameras, one with a 5 m PAN GSD and the other one with a 30 m HS GSD. A forthcoming French mission, HYPXIM/HYPEX-2 [1,2], would ship a 2 m GSD PAN camera and an 8 m GSD HS camera. With such a hyperspectral GSD, accurate urban land cover can be accessible as mentioned by Small [55] who estimated the typical length of urban objects in the 10–20 m range. Nevertheless, at this spatial resolution, the number of mixed pixels remains high. For instance, Wu [56] demonstrated that for a 4 m GSD acquisition with IKONOS data over the town of Grafton, Wisconsin, half of the pixels are mixed pixels. Thus, by combining HS and PAN images, unmixing methods applied on HYPXIM/HYPEX-2 images can significantly improve such urban land cover maps.

The limitations of the current hyperspectral unmixing methods and the perspective of the forthcoming HYPXIM/HYPEX-2 [1,2] mission have motivated the development of a new unmixing approach to reduce those limitations and improve the hyperspectral unmixing process by combining the hyperspectral and panchromatic images. This paper presents an improvement of a strategy to unmix hyperspectral images given an additional PAN image that we originally proposed in Reference [57]. The method is composed of two stages described in Section 2. The first stage called Heterogeneity Based Endmember Extraction (HBEE) extracts a first set of endmembers. The second stage, called Local Constrained Non-negative Matrix Factorization (LCNMF), completes the first set of endmembers with a local iterative approach. The unmixing method is applied, first to a synthetic image to quantify its performance and compare it to a panel of methods. Then, the methods are applied to HYPXIM/HYPEX-2 satellite images simulated from airborne images. In Section 3, synthetic data sets, as well as airborne acquisitions simulated to satellite level images, are presented. The performance criteria, performance assessment procedure and the selection of state-of-the-art methods for comparison are detailed in Section 4. Results are presented and discussed in Section 5. Finally, the main conclusions drawn from this work are given in Section 6.

## 2. Proposed Method

We propose an improved hyperspectral unmixing method, originally introduced in Reference [57], that takes into account a panchromatic image co-registered with a hyperspectral image. This new method, called HBEE-LCNMF, is composed of two stages. The first stage, Heterogeneity-Based Endmember Extraction (HBEE), aims at identifying the endmembers corresponding to pure HS pixels using a heterogeneity criterion applied on the PAN image. This leads to a first set of endmembers that is unlikely complete. Estimating the abundance fractions with this first set of endmembers leads to a rebuilt HS image that has locally high differences with the original HS image. The second stage called Local Constrained Non-negative Matrix Factorization (LCNMF) aims at completing the first set of endmembers extracted by HBEE by applying a sequence of constrained NMFs to the poorly rebuilt areas of the image. As previously mentioned, a standard linear mixture model is considered with the following matrix expression for $N$ HS pixels:

$$Y = XS + W, \tag{3}$$

with $\mathbf{Y} \in \mathbb{R}_+^{N \times N_\lambda}$ the matrix representing the observed data and $\mathbf{W} \in \mathbb{R}^{N \times N_\lambda}$ the noise matrix. $\mathbf{S}$ and $\mathbf{X}$ are, respectively, the endmember matrix and abundance fraction matrix, defined as:

$$
\mathbf{S} = \begin{bmatrix} \vec{s_1}^T \\ \dots \\ \vec{s_p}^T \\ \dots \\ \vec{s_P}^T \end{bmatrix} \in \mathbb{R}_+^{P \times N_\lambda}, \qquad \mathbf{X} = \begin{bmatrix} \vec{x_1}^T \\ \dots \\ \vec{x_n}^T \\ \dots \\ \vec{x_N}^T \end{bmatrix} \in \mathbb{R}_+^{N \times P}, \tag{4}
$$

with $\vec{x_n}$ being the column vector containing the abundance fractions of each endmember for HS pixel $n$.

### 2.1. Heterogeneity-Based Endmember Extraction (HBEE)

The PAN image, thanks to its smaller GSD, provides more detailed spatial information, complementary to the spectral information provided by the hyperspectral image. This leads to multiple PAN pixels in the area covered by a hyperspectral pixel. Using this additional spatial information, and by making the assumption that all the PAN pixels are pure, the goal of the HBEE stage is to detect all the pure HS pixels and then to build a first set of endmembers derived from these pure pixels. This is achieved in three steps. The first step consists in detecting all the pure HS pixels, by assuming that a hyperspectral pixel is pure if the corresponding PAN set of pixels is not heterogeneous (i.e., homogeneous). The spatial PAN heterogeneity can be assessed by a variety of statistical criteria. Instead of the full range criterion on the PAN set of pixels proposed in Reference [57], the 95%–5% percentile range, denoted $\eta$, is chosen to get rid of most of the outliers [58]. With this spatial PAN criterion, a heterogeneity map is computed and all the HS pixels below a $\alpha_h$ threshold are designated as pure. The corresponding spectra are then extracted.

However, several pure pixels can correspond to the same pure material. To keep only one spectral signature for each pure material, a clustering step is then required.

Instead of the clustering procedure proposed in Reference [57], an ascending hierarchical clustering [59] is performed in order to gather the pure spectra. This clustering method is selected as it does not require to know the final number of classes beforehand. Moreover, such a method exhibits good performance to discriminate different classes, knowing the inter-class spectral angle value. At the beginning, each spectrum is put in one distinct class. Then, the algorithm computes a given distance between the representatives of every classes and merges the two closest classes. This merging process is repeated until the distance between the classes to be merged exceeds a given threshold $\alpha_d$. During the clustering process, the representatives $\vec{s_c}$ of each class $c$ are the averages of the spectra of the class weighted by the inverse of the spatial PAN heterogeneity measure related to each observed spectrum (5):

$$
\vec{s_c} = \frac{\sum_{m=1}^{M} \frac{\vec{y_{mc}}}{\eta_{mc} + \epsilon}}{\sum_{m=1}^{M} \frac{1}{\eta_{mc} + \epsilon}}, \tag{5}
$$

with $\vec{y_{mc}}$ the pure spectrum of number $m$ from the class $c$, $M$ the total number of spectra in the class $c$, $\epsilon$ a small positive value, and $\eta_{mc}$ the spatial PAN heterogeneity criterion value of the corresponding HS pixel (the 95%–5% percentile range). This will lead the class representative to be close to its most homogeneous spectrum and then two classes containing similar highly spatially homogeneous spectra will likely be merged. This strategy is a trade-off between choosing the unweighted mean of each class and the most homogeneous spectrum from each class as its representative. The similarity criterion used to merge two classes is the spectral angle measure (SAM) [60] (6) :

$$
SAM(\vec{s_c}, \vec{s_c}') = arccos\left( \frac{\langle \vec{s_c}, \vec{s_c}' \rangle}{\| \vec{s_c} \| . \| \vec{s_c}' \|} \right), \tag{6}
$$

with $\langle , \rangle$ being the scalar product and $||.||$ the vector norm. It has to be noted that, in (6), since the normalization term in Equation (5) is not used, the SAM criterion is no influenced by scale factors on the spectra.

At this point, the number of endmembers represented by pure pixels is derived from the number of classes and then depends on $\alpha_d$ and $\alpha_h$. Finally, the most homogeneous spectra from each class are selected. A first set $\hat{S}_H$ of endmembers, corresponding to materials that have pure HS pixels is thus defined. Nevertheless, it is likely that endmembers which are present only in mixed pixels exist and thus have not been detected and extracted by this process. This is the goal of the next stage.

### 2.2. Local Constrained Non-Negative Matrix Factorization (LCNMF)

The second stage of this unmixing process is called Local Constrained Non-negative Matrix Factorization (LCNMF). Its goal is to complete the first set $\hat{S}_H$ of endmembers by extracting from mixed pixels the remaining $P_m$ endmembers applying a sequence of constrained NMFs to the HS image. This method is an improved version of the local iterative algorithm proposed in Reference [57] called Local semi-Supervised Non-negative Matrix Factorization, the main improvements being the use of constraints and a different initialization procedure. The algorithm is composed of two iteratively repeated parts.

#### 2.2.1. Partial Unmixing Step and Reconstruction Error

The algorithm applies a sequence of constrained NMFs to $K$ areas of interest, with the current NMF run and area being indexed by the subscript $k$. The areas of interest which correspond to high reconstruction errors computed after the run $k - 1$, are supposed to have mixed pixels in which spectra contain the spectral contribution of one additional (yet) unknown endmember. The detection of such areas in the image is explained hereafter. Until the algorithm stops, the matrix $\hat{S}_k$, which holds the known endmember spectra at run $k$, does not contain all the endmembers present in the image. At run $k = 0$, the matrix $\hat{S}_k$ is initialized with $\hat{S}_H$. Consequently, using only $\hat{S}_k$ to estimate the abundance fractions induces that the ASC does not hold on the entire image. In this regard, we use the NNLS algorithm [61], which does not account for the ASC, to estimate $\hat{X}_{N,k}$ and rebuild the image $\hat{Y}_{N,k}$ with:

$$\hat{Y}_{N,k} = \hat{X}_{N,k}\hat{S}_k, \tag{7}$$

with $\hat{Y}_{N,k} \in \mathbb{R}_+^{N \times N_\lambda}$ the $N$ as rebuilt pixels of the HS image at run $k$, $\hat{S}_k \in \mathbb{R}_+^{P(k) \times N_\lambda}$ as the matrix containing the set of endmembers at run $k$, and $\hat{X}_{N,k} \in \mathbb{R}_+^{N \times P(k)}$ as the abundance fraction matrix for all the $N$ pixels of the HS image at run $k$. The matrices $\hat{X}_{N,k}$ and $\hat{Y}_{N,k}$ should not be confused with $\hat{X}_k$ and $\hat{Y}_k$; their notations will be introduced later. Finally, the reconstruction error is calculated for each pixel $n$ using (8):

$$r_{nk} = \frac{||\overrightarrow{y_n} - \overrightarrow{\hat{y}_{nk}}||}{||\overrightarrow{y_n}||}, \tag{8}$$

leading to the Reconstruction Error Map (REM) at run $k$.

#### 2.2.2. Iterative Process to Estimate Unknown Endmembers

After the unmixing step carried out with the endmember set obtained by HBEE (first run), and then for each run $k$, the REM is computed. In presence of endmembers not yet extracted after iteration $k$, local areas, which are the areas of interest, exhibit high reconstruction errors. The presence of only one unknown endmember in each poorly rebuilt area is assumed as previously mentioned. To detect these poorly rebuilt contiguous areas, a relative threshold $\alpha_{r(k)}$ is first calculated according to:

$$\alpha_{r(k)} = percentile_{95}(r_{nk}), \tag{9}$$

with *percentile*$_{95}$ being the value separating the 95% lowest values of the data from the 5% highest. Then, this threshold value is applied on the REM to highlight the 5% pixels with the highest reconstruction error. These pixels form several contiguous areas which are distinguished with an adequate image processing algorithm, e.g., the label function from the scipy.ndimage python library (documentation of the scipy.ndimage.label python function: https://docs.scipy.org/doc/scipy/reference/generated/scipy.ndimage.label.html).

The area containing the poorest rebuilt spectrum is selected as the current HS area. This area defines the current set $Y_k$ composed of $N_k$ observed spectra from $Y$, as well as the local abundance fraction matrix $\hat{X}_k \in \mathbb{R}_+^{N_k \times P(k)}$. $N$ and $N_k$ should not be confused as $N$ represents the number of HS pixels in the entire image, and $N_k$ defines the number of HS pixels in the area of interest number $k$. Safeguard parameters may be useful to limit the minimum or maximum size of the detected areas. The maximum size ensures that only one unknown endmember is present and the minimum size is set to have a sufficient amount of data to process. In our case, if the area contains only one pixel $n$, the NMF is applied on that pixel, and its 8 neighbours, that form the observed spectra subset $Y_k$. This strategy naturally leads the NMF being applied to a quite small number of HS pixels (depending on the threshold $\alpha_{r(k)}$), reducing the spectral variability of the material for which the endmember is sought. The assumption of the presence of only one unknown endmember leads to the extension of the endmember and abundance matrices $\hat{S}_k$ and $\hat{X}_k$ by, respectively, one row and one column to account for the unknown endmember to be estimated, meaning $\hat{P}_k \leftarrow \hat{P}_{k-1} + 1$.

The matrix $\hat{S}_k$ is initialized with the endmembers already estimated by (1) HBEE and (2) the $k-1$ previous iterations and the spectrum $\overrightarrow{y_{maxerror}}$ from $Y_k$ corresponding to the highest reconstruction error in the considered area:

$$\hat{S}_{k,init} = \begin{pmatrix} \hat{S}_{k-1} \\ \overrightarrow{y_{maxerror}} \end{pmatrix}. \tag{10}$$

The initial local abundance matrix $\hat{X}_k$ is obtained by using the Fully-constrained Least Square (FCLS) algorithm [62] with $Y_k$ and $\hat{S}_k$.

Then, a constrained NMF, which is described further in this paper, is performed to estimate the unknown endmember located in the area. The matrix $\hat{S}_k$ is finally saved and the algorithm steps forward to the next run $k \leftarrow k + 1$. The overall procedure is repeated until all the reconstructed pixels show a reconstruction error (RE) lower than a given threshold $\alpha_{RE}$ which might be set close to the residual error background. At the end of the overall process, the complete endmember matrix $\hat{S}$ describing all the $P$ materials in the scene has been estimated. The final abundance fraction matrix $\hat{X}$ is estimated using FCLS. The overall procedure is summarized in Algorithm 1.

This approach does require a minimum amount of known material signatures to begin its process. Otherwise, the number of poorly reconstructed areas would be too high and they could overlap, creating larger poorly rebuilt areas. Therefore the assumption of a single unknown endmember in each poorly rebuilt area would not hold. Nevertheless, the procedure estimates the number of endmembers alongside its iterations and thus, does not require it to be known and set beforehand.

---

**Algorithm 1:** The LCNMF procedure.

---

**Input data:** $Y \in \mathbb{R}_+^{N \times N_\lambda}$, $\hat{S}_H \in \mathbb{R}_+^{P_H \times N_\lambda}$

**Input parameters:** $\alpha_{RE}$

1　$k = 0$
2　$\hat{S}_k \leftarrow \hat{S}_H$
3　$\hat{X}_{N,k} \leftarrow NNLS(Y, \hat{S}_k)$
4　$\hat{Y}_{N,k} \leftarrow \hat{X}_{N,k} \hat{S}_k$
5　**forall** *pixels n* **do**
6　$\quad \left| \quad r_{nk} \leftarrow \frac{||\vec{y_n} - \vec{\hat{y}_{nk}}||}{||\vec{y_n}||} \right.$
　　**end**
7　$k \leftarrow k + 1$
8　**while** *at least one pixel shows a RE error higher than* $\alpha_{RE}$ **do**
9　$\quad$ Calculate $\alpha_{r_{nk}}$ using (9).
10　$\quad$ Threshold the REM using $\alpha_{r_{nk}}$ (9).
11　$\quad$ Select the area containing the highest reconstruction error value as the current area.
12　$\quad$ Initialization of $\hat{S}_k$ using Equation (10).
13　$\quad \hat{X}_{k,init} \leftarrow FCLS(Y_k, \hat{S}_{k,init})$
14　$\quad \hat{S}_k \leftarrow$ constrained $NMF(Y_k, \hat{S}_{k,init}, \hat{X}_{k,init})$
15　$\quad \hat{X}_{N,k} \leftarrow NNLS(Y, \hat{S}_k)$
16　$\quad \hat{Y}_{N,k} \leftarrow \hat{X}_{N,k} \hat{S}_k$
17　$\quad$ **forall** *pixels n* **do**
18　$\quad \quad \left| \quad r_{nk} \leftarrow \frac{||\vec{y_n} - \vec{\hat{y}_{nk}}||}{||\vec{y_n}||} \right.$
　　$\quad$ **end**
19　$\quad k \leftarrow k + 1$
　　**end**
20　$\hat{S} \leftarrow \hat{S}_k$
　　**Result:** $\hat{S} \in \mathbb{R}_+^{P \times N_\lambda}$

---

### 2.2.3. Constrained NMF

The small number of HS pixels in the considered area and the assumed presence of only one endmember to be estimated induce two constraints. First, even if endmembers are missing, the number of endmembers present in the considered poorly rebuilt areas must be lower than or equal to the number of known materials plus one, the unknown endmember to be estimated. Then, the ASC can be applied. Secondly, the endmembers extracted at run number $k$ are not expected to vary at run number $k + 1$. It is therefore necessary to prevent their variation in the NMF process at run number $k + 1$. These constraints can easily be taken in account in the NMF, which is why this estimating framework has been chosen.

To fulfill the ASC, the strategy proposed with the FCLSU method in Reference [62] is used. Column vectors $\vec{1} = [1, \dots, 1]^T \in \mathbb{R}_+^{N_k}$ and $\vec{1} = [1, \dots, 1]^T \in \mathbb{R}_+^{P(k)}$ are, respectively, appended to the matrices $Y_k$ and $\hat{S}_k$. The second constraint aims at blocking the variations of endmembers extracted during the previous runs, by using the strategy proposed by Karoui et al. [63] with the corresponding parameter $L_2 = 1$. The matrix $\hat{S}_k$ is separated in two matrices:

$$\hat{S}_k = S_{k1} + \hat{S}_{k2} \tag{11}$$

with:

$$S_{k1} = \begin{pmatrix} \vec{s_1} \\ \vdots \\ \vec{s}_{P(k)-1} \\ \vec{0}_{N_\lambda}^T \end{pmatrix} \in \mathbb{R}_+^{P(k)\times(N_\lambda+1)} \qquad \hat{S}_{k2} = \begin{pmatrix} \vec{0}_{N_\lambda}^T \\ \vdots \\ \vec{0}_{N_\lambda}^T \\ \hat{\vec{s}}_{P(k)} \end{pmatrix} \in \mathbb{R}_+^{P(k)\times(N_\lambda+1)}. \tag{12}$$

This leads to Equation (13):

$$\hat{Y}_k = \hat{X}_k \left( S_{k1} + \hat{S}_{k2} \right). \tag{13}$$

The objective function $J(\hat{X}_k, \hat{S}_k)$ is the well known reconstruction error given in (14):

$$J(\hat{X}_k, \hat{S}_{k2}) = ||Y_k - \hat{X}_k \left( S_{k1} + \hat{S}_{k2} \right)||_F^2, \tag{14}$$

$$= tr\left(Y_k Y_k^T\right) - 2tr\left(Y_k S_{k1}^T \hat{X}_k^T\right) - 2tr\left(Y_k \hat{S}_{k2}^T \hat{X}_k^T\right) + tr\left(\hat{X}_k S_{k1} S_{k1}^T \hat{X}_k^T\right) \tag{15}$$

$$+ 2tr\left(\hat{X}_k S_{k1} \hat{S}_{k2}^T \hat{X}_k^T\right) + tr\left(\hat{X}_k \hat{S}_{k2} \hat{S}_{k2}^T \hat{X}_k^T\right),$$

with $tr()$ being the trace of a matrix. To obtain the projected gradient additive update rules, the gradients of Equation (16) are calculated with respect to $\hat{X}_k$ and $\hat{S}_{k2}$. In our case, the expressions of the gradients are:

$$\frac{\partial J(\hat{X}_k, \hat{S}_{k2})}{\partial \hat{X}_k} = 2\hat{X}_k \left[ \left(S_{k1} + \hat{S}_{k2}\right)\left(S_{k1} + \hat{S}_{k2}\right)^T \right] - 2Y_k \left(S_{k1} + \hat{S}_{k2}\right)^T, \tag{16}$$

$$\frac{\partial J(\hat{X}_k, \hat{S}_{k2})}{\partial \hat{S}_{k2}} = 2\hat{X}_k^T \hat{X}_k \left(S_{k1} + \hat{S}_{k2}\right) - 2\hat{X}_k^T Y_k. \tag{17}$$

The projected gradient additive update rules are (18), (19):

$$\hat{X}_k \leftarrow max\left(\hat{X}_k - \beta_{\hat{X}_k} \odot \frac{\partial J(\hat{X}_k, \hat{S}_{k2})}{\partial \hat{X}_k}, \epsilon\right), \tag{18}$$

$$\hat{S}_{k2} \leftarrow max\left(\hat{S}_{k2} - \beta_{\hat{S}_{k2}} \odot \frac{\partial J(\hat{X}_k, \hat{S}_{k2})}{\partial \hat{S}_{k2}}, \epsilon\right), \tag{19}$$

with $\beta_{\hat{X}_k}$ and $\beta_{\hat{S}_{k2}}$, the matrix-form gradient steps, $\odot$ the component-wise product, and $\epsilon$ a positive value close to zero.

The fixed-step projected gradient additive update rules have two drawbacks. First, a relatively small value must be chosen for $\beta_{\hat{X}_k}$ and $\beta_{\hat{S}_{k2}}$ [34,64] and will influence the convergence. For a too large value, the convergence is not guaranteed and for a too small value the convergence may be extremely slow. Advanced steepest projected gradient algorithms with adaptive learning steps to speed-up convergence are described in Reference [65], but the computational time is not a parameter we wanted to optimize. Thus, this method has therefore not been selected. Secondly, the non-negativity of the result is not guaranteed and a projection in $\mathbb{R}_+$ is needed. For these reasons, multiplicative update rules have been chosen. The procedure to derive multiplicative-update rules from gradient-based additive update rules has been described in Reference [63,66]. It consists of expressing the gradient steps as:

$$\beta_{\hat{X}_k} = \hat{X}_k \oslash \left(2\hat{X}_k \left[\left(S_{k1} + \hat{S}_{k2}\right)\left(S_{k1} + \hat{S}_{k2}\right)^T\right]\right), \tag{20}$$

$$\beta_{\hat{S}_{k2}} = \hat{S}_{k2} \oslash \left(2\hat{X}_k^T \hat{X}_k \left(S_{k1} + \hat{S}_{k2}\right)\right), \tag{21}$$

with $\oslash$ the component-wise division. The resulting multiplicative update rules become:

$$\hat{X}_k \leftarrow \hat{X}_k \odot \left( Y_k \left( S_{k1} + \hat{S}_{k2} \right)^T \right) \oslash \left( \hat{X}_k \left( S_{k1} + \hat{S}_{k2} \right) \left( S_{k1} + \hat{S}_{k2} \right)^T + \epsilon \right), \tag{22}$$

$$\hat{S}_{k2} \leftarrow \hat{S}_{k2} \odot \left( \hat{X}_k^T Y_k \right) \oslash \left( \hat{X}_k^T \hat{X}_k \left( S_{k1} + \hat{S}_{k2} \right) + \epsilon \right). \tag{23}$$

The constrained NMF ends when the objective function $J(\hat{X}_k, \hat{S}_k)$ becomes lower than a stopping error threshold $\alpha_{stop}$ or if the number of iterations exceeds a given maximum number of iterations.

### 2.3. HBEE-LCNMF Overview

Finally, the overall proposed method can be summarized in Figure 2.

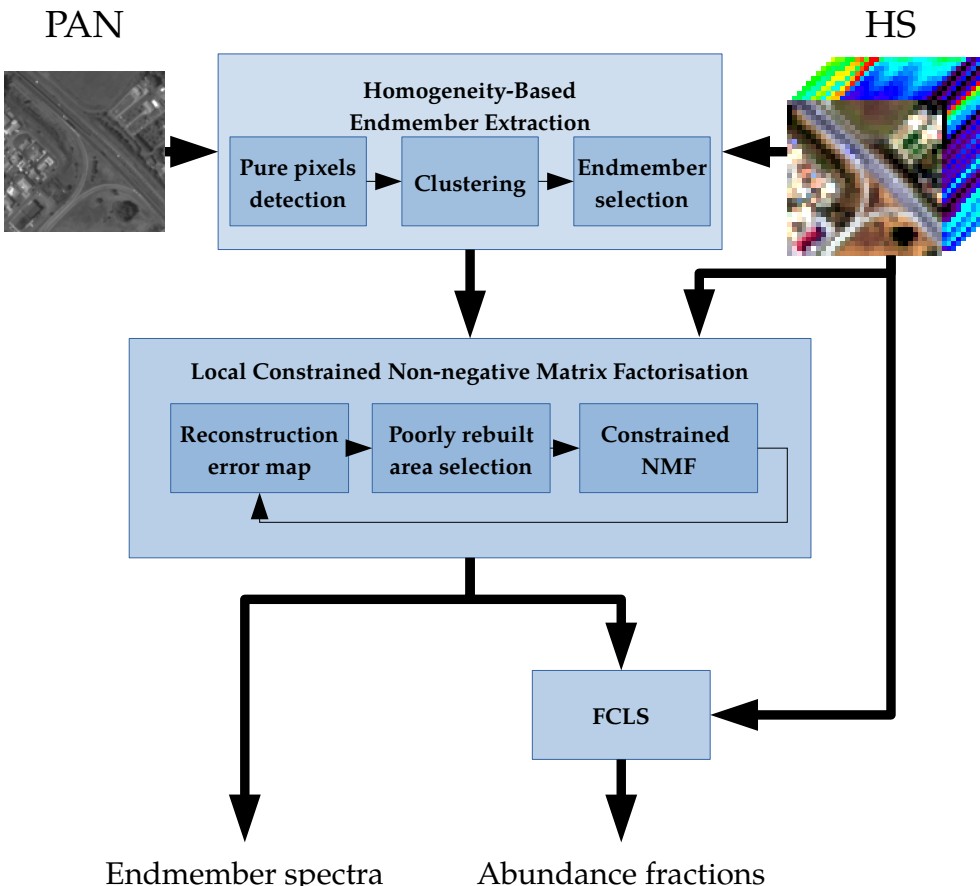

**Figure 2.** Overview of the proposed Local Constrained Non-negative Matrix Factorization (HBEE-LCNMF) method.

## 3. Materials

In order to assess the performance of the proposed approach, several data sets are considered. First, a synthetic data set has been generated using pure material reflectance spectra. Secondly, two small airborne HS image subsets have been selected, their composition is known in terms of the number and type of different materials present in the scenes.

From these data sets, satellite images have been simulated considering the HYPXIM/HYPEX-2 spatial and spectral characteristics. The PAN image is in radiance units at the satellite level, and the hyperspectral image is in reflectance units at ground level, after applying atmospheric compensation.

### 3.1. Synthetic Data Set

A synthetic data set is considered in order to assess the behavior and robustness of the proposed approach when its assumptions are fully verified. To this end, a 2 m GSD class image, shown in Figure 3, that mimics a quite realistic spatial layout is created. A 2 m GSD ground reflectance HS image is then synthesized by placing in each pixel a spectrum randomly selected from a set of spectra corresponding to the material class assigned to the pixel. In this way, we ensure a quite realistic spectral variability in each class. The reference material spectra have been extracted from homogeneous areas of a 2 m GSD ground reflectance HS image acquired with the APEX sensor over Basel (Switzerland), except for the tennis red surface material, which has been acquired with the ODIN sensor over Canjuers (France). These spectra have been resampled to the HYPXIM wavelengths and are shown in Figure 4. The obtained image is downsampled to 8 m GSD considering a square Point Spread Function (PSF) on the surface spanned by the 8 m GSD HS pixels.

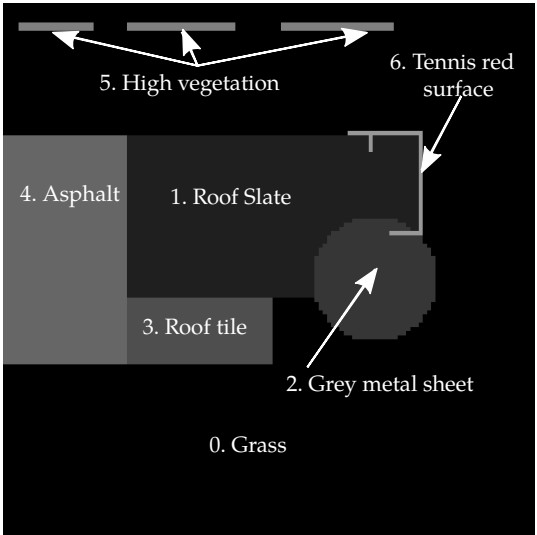

**Figure 3.** Class image at 2 m Ground Sampling Distance (GSD) with seven distinct materials. The area spanned by the high vegetation is 4 m wide. The tennis red surface is 2 m wide. At the hyperspectral (HS) image resolution, those two materials are present only in mixed pixels.

A noise component, according to the HYPXIM noise model provided by CNES (Center National d'Études Spatiales, France) has been added with an SNR of 100 (20 dB) for the 0.5 μm spectral band. The noise model, for a given Top Of Atmosphere (TOA) radiance pixel, can be written as follows:

$$w_{n,L}(\lambda) = q\sqrt{a(\lambda) + b(\lambda)L_n(\lambda)} \qquad q \sim \mathcal{N}(0,1), \qquad (24)$$

where $a(\lambda)$ in $(W/m^2/sr/\mu m)^2$ and $b(\lambda)$ in $W/m^2/sr/\mu m$. $L_n(\lambda)$ is the radiance at the sensor input in $W/m^2/sr/\mu m$. For the sake of simplicity, we considered an approximate reflectance equivalent noise model defined as:

$$w_{n,\rho}(\lambda) = q\frac{\pi}{E_t(\lambda)\tau_t(\lambda)}\sqrt{a(\lambda) + b(\lambda)\left(L_{atm}(\lambda) + \frac{E_t(\lambda)\tau_t(\lambda)}{\pi}\rho_n(\lambda)\right)} \qquad q \sim \mathcal{N}(0,1), \qquad (25)$$

with $L_{atm}(\lambda)$ as the up-welling atmospheric radiance in $W.m^{-2}.sr^{-1}.\mu m^{-1}$, $E_t(\lambda)$ the total irradiance over the considered ground surface in $W.m^{-2}.\mu m^{-1}$, $\tau_t(\lambda)$ the transmission coefficient between the ground surface and the sensor, and $\rho_n(\lambda)$ the ground reflectance of pixel $n$. The radiative parameters have been calculated using the COMANCHE [67] code. The 8 m GSD ground reflectance image is then spectrally smoothed using a Gaussian kernel with five components. This smoothing procedure is done in order to attenuate the previously applied noise as it would be done on a real image before processing.

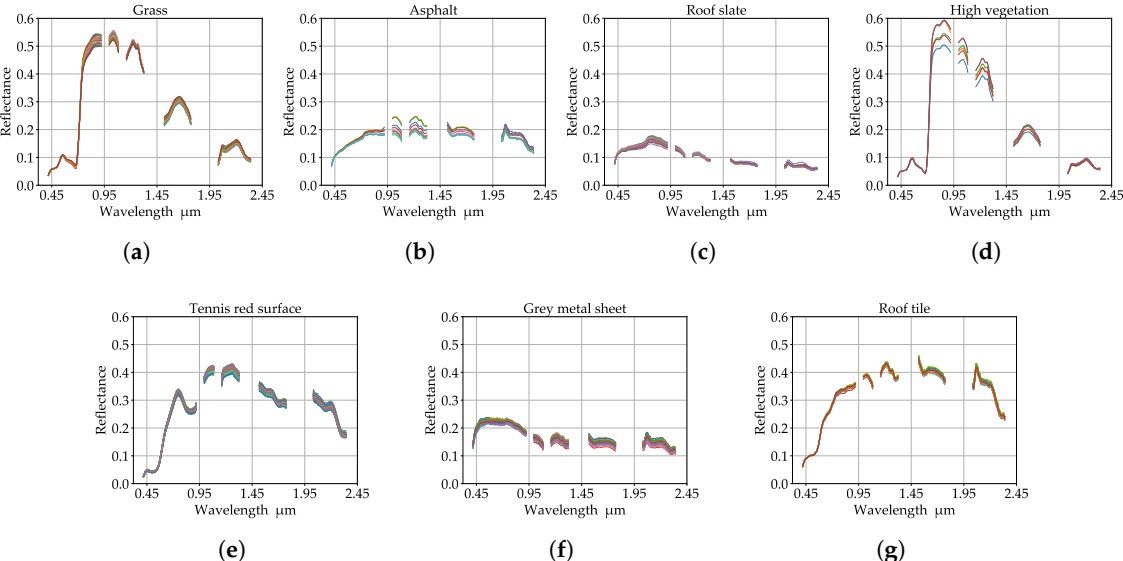

**Figure 4.** Reference spectra used to build the synthetic images. The material classes are: (**a**) grass, (**b**) asphalt, (**c**) roof slate, (**d**) high vegetation, (**e**) tennis red surface, (**f**) grey metal sheet and (**g**) roof tile.

The simulation of the 2 m GSD Top-Of-Atmosphere PAN image is done as follows: first, an atmospheric model, using COMANCHE, is applied on the 2 m GSD ground reflectance HS image in order to have a 2 m GSD TOA radiance HS image. Secondly, a HS noise component, considering the HYPXIM noise model is then added using Equation (24). Finally, a spectral binning using a square window from 0.4 μm to 0.8 μm is performed to obtain the 2 m GSD PAN image. The resulting synthetic data set is shown in Figure 5.

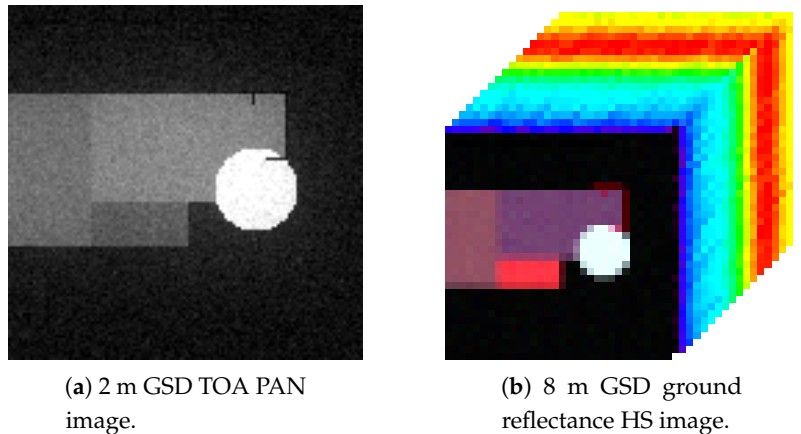

(**a**) 2 m GSD TOA PAN image.

(**b**) 8 m GSD ground reflectance HS image.

**Figure 5.** Synthetic data set.

*3.2. Satellite Simulated Airborne Data Sets*

Although synthetic data allow us to assess the main hypothesis of the proposed approach, such data do not account for real acquisition features, such as shadows, real spectral variability, and realistic spatial distribution of the materials, in the scene, even if these aspects, except shadows, have been taken into account as much as possible in the construction of our synthetic image. In this regard, satellite simulation from airborne acquisitions have been considered, as well. The main difficulty encountered with real data is often related to the lack of ground truth. To tackle this issue, we selected sufficiently small areas in two airborne HS images with GSDs of 55 cm and 2 m containing a known number of endmembers. Then, thanks to the low GSD, pure spectra for each material are

extracted, to which the retrieved endmembers will be compared. The first image has been acquired over the village of Mauzac near Toulouse (France) with the Aisa-Fenix sensor (see Figure 6a). The GSD is 55 cm and the scene is composed of five distinct materials, and their reflectance spectra are drawn in Figure 7. The second image has been acquired over the town of Basel in Switzerland with the APEX sensor [68] (see Figure 6b). The GSD is 2 m and the image is composed of seven materials in which their reflectance spectra are shown in Figure 8. One can notice the similarity between some spectra in each image, especially in the Short Wave InfraRed (SWIR) spectral range ([1.0, 2.5] µm), where some spectral features are shared by all synthetic materials. This may affect the unmixing process as the vertices of the simplex induced by the data might be close to each other. Moreover the green fence and the green track materials do not completely fill a pixel because they are about 1.5 m in their smallest dimension. In the 2 m GSD image, we can therefore estimate at 56% the maximum percentage of these materials in the few pixels where they are present. Consequently they are highly mixed in the 8 m GSD image. The extraction of their spectral signatures from this image is therefore very challenging.

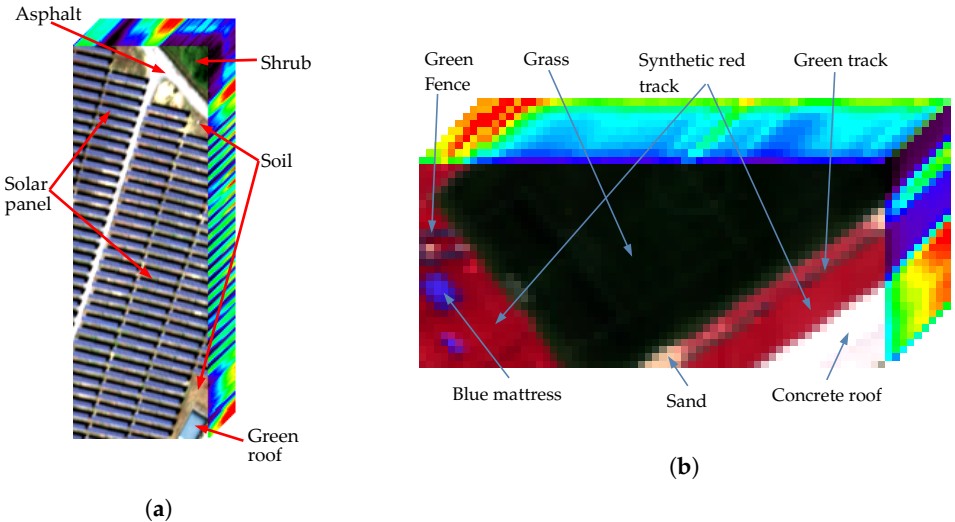

**Figure 6.** Airborne HS images: (**a**) 55 cm GSD Mauzac HS image, (**b**) 2 m GSD Basel HS image.

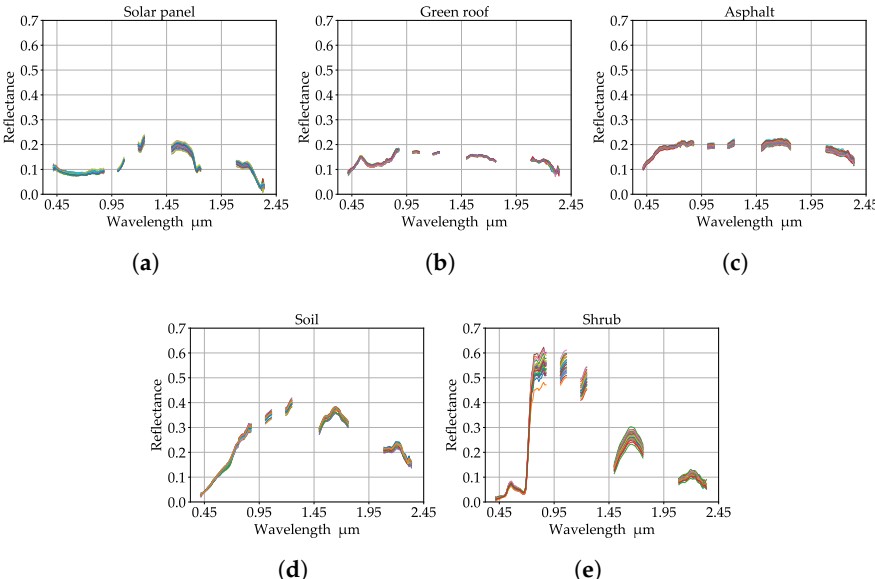

**Figure 7.** Pure spectra extracted from the Mauzac image for each class of materials. The material classes are: (**a**) solar panel, (**b**) green roof, (**c**) asphalt, (**d**) soil, (**e**) shrub.

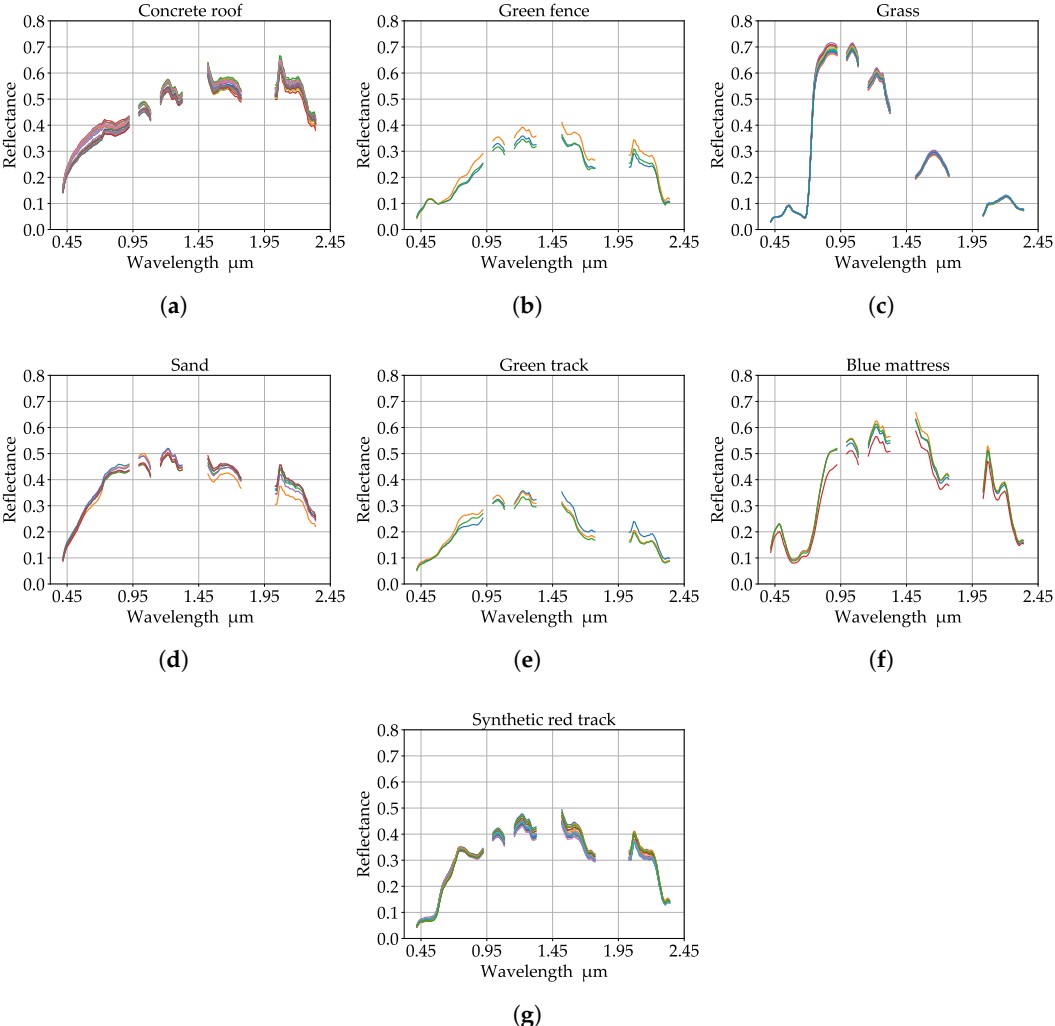

**Figure 8.** Pure spectra extracted from the Basel image for each class of materials. The material classes are: (**a**) concrete roof, (**b**) green fence, (**c**) grass, (**d**) sand, (**e**) green track, (**f**) blue mattress, (**g**) synthetic red track.

HYPXIM satellite images have been simulated using the procedure described in Section 3.1, except that the HS spatial binning has been performed using the PSF of the HS HYPXIM instrument provided by CNES. For the Mauzac image, the spatial binning used in the PAN image simulation has been performed according to a spatial square window. The two resulting data sets are shown in Figure 9.

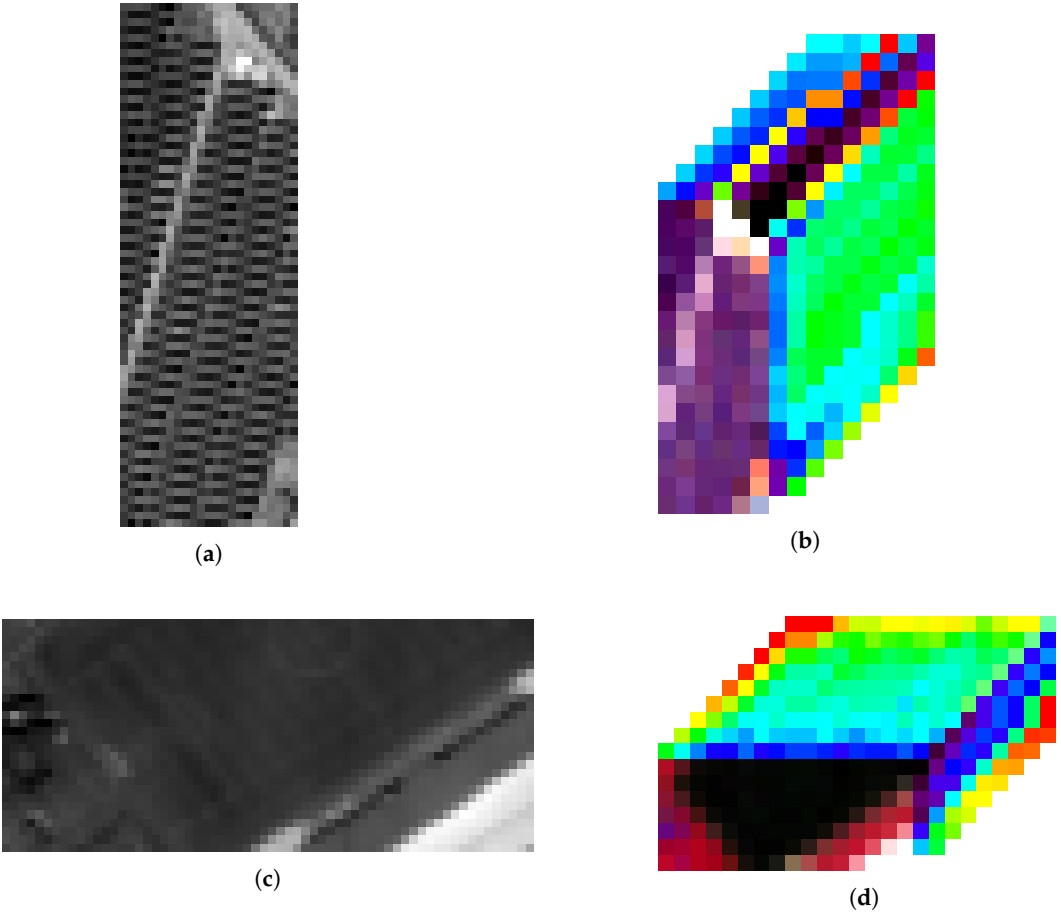

**Figure 9.** HYPXIM/HYPEX2 simulated data sets: (**a**) Mauzac 2.2 m GSD TOA PAN image, (**b**) Mauzac 8.8 m GSD ground reflectance HS image, (**c**) Basel 2 m GSD TOA PAN image and (**d**) Basel 8 m GSD ground reflectance HS image.

## 4. Performance Assessment Protocol

In this section, performance criteria are chosen to evaluate the unmixing methods and a procedure is proposed to assess their global performance. A set of state-of-the-art methods are also chosen for comparison. Finally, a protocol to set the HBEE-LCNMF parameters is proposed and the parameters of the state of the art methods are given.

### 4.1. Performance Criteria

To assess the spectral performance of our unmixing method, as well as those that are state-of-the-art, four performance criteria are calculated between the estimated endmembers and the spectral mean of each reference class. The first performance criterion is the spectral angle (see Equation (6)) which compares the shapes of two spectra. The second one is the Normalized Root Mean Square Error (NRMSE) (26), which evaluates the normalized root mean square difference of reflectance levels between two spectra:

$$NRMSE(\vec{s_p}, \vec{\hat{s}_{p'}}) = \frac{||\vec{s_p} - \vec{\hat{s}_{p'}}||}{||\vec{s_p}||}, \tag{26}$$

with $\vec{s_p}$ and $\vec{\hat{s}_{p'}}$ being, respectively, the reference spectrum number $p$ and the estimated endmember spectrum number $p'$ sampled on $N_\lambda$ spectral bands.

The third performance criterion is the Spectral Information Divergence (SID) [69] measuring the spectral similarity of two spectra:

$$SID(\vec{s_p}, \vec{\hat{s}_{p'}}) = D\left(\vec{s_p} || \vec{\hat{s}_{p'}}\right) + D\left(\vec{\hat{s}_{p'}} || \vec{s_p}\right), \tag{27}$$

where $D\left(\vec{x} || \vec{y}\right)$ is the Kullback-Leibler divergence [70].

The last one is the Root Mean Square Error (RMSE) (28), which evaluates the root mean square difference of reflectance levels between two spectra:

$$RMSE(\vec{s_p}, \vec{\hat{s}_{p'}}) = ||\vec{s_p} - \vec{\hat{s}_{p'}}||. \tag{28}$$

These criteria are computed between each estimated endmember and each spectral mean of the reference classes of materials, thus forming a score table for each criterion. To match each estimated endmember with its corresponding reference class and to get the global performance, the following iterative procedure has been applied. We select the best value in the table and match the corresponding estimated endmember and reference. The matched endmember and reference are then removed from the matching procedure. The procedure stops when each of the endmembers is matched with a reference.

To assess the abundance estimation accuracy of each method, the NRMSE has been used:

$$NRMSE(\vec{x_p}, \vec{\hat{x}_{p'}}) = \frac{||\vec{x_p} - \vec{\hat{x}_{p'}}||}{||\vec{x_p}||}, \tag{29}$$

with $\vec{x_p} \in \mathbb{R}_+^N$, the reference abundance fractions for all pixels and for the real endmember number $p$, $\vec{\hat{x}_{p'}} \in \mathbb{R}_+^N$, the estimated abundance fractions for all pixels and for endmember number $p'$. The RMSE has been considered, as well:

$$RMSE(\vec{x_p}, \vec{\hat{x}_{p'}}) = ||\vec{x_p} - \vec{\hat{x}_{p'}}||. \tag{30}$$

The estimated and reference abundance fractions are compared according to the estimated-reference spectra couples previously paired according to the SAM criterion. For methods that estimate only the endmembers, abundance fractions are retrieved with the FCLS method.

## 4.2. State-Of-The-Art Methods Chosen for Comparison

Seven unmixing methods have been selected from all the method families in order to confront all the unmixing frameworks to the proposed approach. Although these methods are not the most recently published, they provide a good overview of what can be expected from state-of-the-art unmixing performance and most of them have been extensively used by the scientific community. These methods are: VCA (VCA python code: https://github.com/Laadr/VCA) [18], ATGP (ATGP python code: https://pysptools.sourceforge.io/_modules/pysptools/eea/eea.html#ATGP) [20], and N-FINDR (N-FINDR python code: https://pysptools.sourceforge.io/_modules/pysptools/eea/eea_int.html#NFINDR) [19] for the pure-pixel detection-based methods, SISAL (SISAL python code: https://github.com/etienne-monier/lib-unmixing) [23] for the purely minimum-volume based methods, MVCNMF (MVCNMF python code: https://github.com/bm424/mvcnmf) [26] for the NMF-based methods, and JBEELU (JBEELU MATLAB code: http://dobigeon.perso.enseeiht.fr/applications/app_hyper_SMA.html) [24] for the Bayesian methods. In addition, the previous version of the proposed approach, Homogeneity-Based Endmember Extraction—Local Semi-supervised Non-negative Matrix Factorization (HBEE-LSNMF), described in Reference [57], is also confronted by HBEE-LCNMF.

### 4.3. Parameter Setting

The HBEE parameters $\alpha_h$ and $\alpha_d$ have been set regarding the underlying hypothesis of the dissimilarity between classes of pure materials. $\alpha_h$ is first set to detect a reasonably small number of pure pixels. $\alpha_d$ is then set step by step to have the most homogeneous classes in the clustering step of HBEE with the help of a human expert. These classes also must be distinguishable from each other. If this is not possible, it likely means that non-pure spectra have been considered as pure spectra, then the value of $\alpha_h$ must be reduced.

For the LCNMF parameters, the $\alpha_{RE}$ threshold is roughly fixed by the user regarding the reconstruction error map background value. Such a value may be estimated by visually selecting a consistent number of background error pixels on the REM and deducing their mean error. Estimating spectra that are close to previously estimated spectra likely means that this threshold induced the selection of an area with an error lower than the background error. Then, $\alpha_{RE}$ must be increased. Thanks to the reduced computational cost of each NMF (only a few pixels are processed), the stopping criterion of the constrained NMFs is set to a high number of iterations, typically 10,000.

The MVCNMF method has been applied with the parameter values (defined in Reference [26]) given in Table 1.

**Table 1.** MVCNMF parameter values.

|  | Fit_Tolerance | Convergence_Tolerance | Learning_Rate | Learning_Tolerance | Max_Iter |
|---|---|---|---|---|---|
| MVCNMF | $10^{-5}$ | $10^{-4}$ | $10^{-2}$ | $10^{-5}$ | $10^3$ |

The parameters of the other methods have been set to recommended values except the regularization parameter $\mu$ of the SISAL method set to 0.1 to ensure the convergence. The non pure-pixel-based methods, i.e., SISAL, MVCNMF, and JBEELU, are initialized with the VCA outputs. For commodity purposes, N-FINDR is initialized with the ATGP algorithm, which only affects the computational time and not the final result of the method.

Similarly to HBEE-LCNMF, the HBEE-LSNMF parameters are the maximum-minimum heterogeneity threshold $t_h$, the classification spectral angle threshold $t_{sa}$, the reconstruction error threshold $t_{RE}$, the stopping NMF error value, the NMF maximum number of iterations and $a$, the dividing coefficient to initialize the local abundance fraction. Their values will be described for each data set, in the corresponding result sections.

## 5. Results and Discussion

This section compares the performances of HBEE-LCNMF and of the selected state-of- the-art methods.

First the unmixing results on the synthetic image are analyzed to validate the proposed approach. Finally, the results of the unmixing methods on Bale and Mauzac images derived from real images are shown and discussed.

### 5.1. Unmixing Results for the Synthetic Data Set

The HBEE-LCNMF method has been applied with the parameters given in Table 2. The number of endmembers determined from the ground truth is 7. The HBEE-LCNMF method has been applied with the parameters given in Table 3.

**Table 2.** HBEE-LCNMF parameter values for the synthetic data set.

|  | $\alpha_h$ | $\alpha_d$ | $\alpha_{RE}$ | Stopping NMF Error Value | NMF Max Iteration |
|---|---|---|---|---|---|
| value | $2.2\,\text{W/m}^2/\text{sr}$ | $5.4°$ | $5 \times 10^{-2}$ | $5 \times 10^{-8}$ | $10^5$ |

**Table 3.** HBEE-LSNMF parameter values for the synthetic data set.

|  | $t_h$ | $t_{sa}$ | $t_{RE}$ | Stopping NMF Error Value | NMF Max Iteration | a |
|---|---|---|---|---|---|---|
| value | 3 W/m$^2$/sr | 7° | 0.05 | $10^{-7}$ | $10^5$ | 3 |

HBEE-LCNMF estimates all 7 endmembers. The other methods have been applied to estimate 7 endmembers. It can be noted that the HYSIME method estimates this number at 20, whereas HFC estimates it at 6, 5, 5 according to the value of the false alarm parameter, respectively, set at $10^{-3}$, then $10^{-4}$ and $10^{-5}$. The performance of the HBEE stage is equivalent to that of pure pixel-based methods for endmembers represented by pure pixels. This can be seen in Figure 10 as the spectra which have been estimated from pure pixels (a, b, c, d, e) by HBEE are quite close to the reference spectra. The main contribution of the HBEE stage is, as expected, to estimate the number of materials represented by pure pixels which none of the state-of-the-art methods can do.

Complementary to the results of HBEE, the two highly mixed endmembers, i.e., the high vegetation and the tennis red surface material, are quite well estimated by LCNMF considering their low contribution in the mixed pixels, especially for the Tennis red surface endmember. While it is not surprising that pure-pixel based methods fail to extract these endmembers present in small proportions in the pixels, it can be noticed that the other methods also fail to extract the tennis red surface material. One can highlight the fact that the previous HBEE-LSNMF method does not process areas that are composed of less than 3 pixels. Therefore, the LSNMF stage does not detect the area containing the tennis red surface material. Moreover, The high vegetation material is not correctly estimated, which maintains a high reconstruction error in other areas where this material is located. The procedure then selects one of these other areas and performs the classical NMF on it to estimate an endmember which has already been estimated. This results in a redundant estimate. As we can see in Figure 11, the radar plots of HBEE-LCNMF are enclosed within the plots of the other unmixing methods. This means that HBEE-LCNMF estimates are closer to the reference spectra than the ones of the other methods. The resulting mean criterion values, given in Table 4, show an improvement of the normalized difference by 51% in SAM ($\frac{3.9-1.9}{3.9} \times 100$), 85% in SID, 55% in RMSE, and 48% in NRMSE using HBEE-LCNMF compared to the best state-of-the-art method (VCA).

The abundance fractions shown in Figure 12 are accurately estimated by our method since the correct locations are retrieved whatever the endmember. The compared performances between the estimated and reference abundance fractions show an improvement of 44% compared to the best state-of-the-art methods (VCA).

This test on synthetic data is conclusive because our method behaves as expected and it avoids the common limitations that are due to the absence of pure pixels for some endmembers and the non-convexity of the problem addressed by non-pure pixel based algorithms. Moreover, the computational time of the method is equivalent to that of MVCNMF and is acceptable. Nevertheless, although we have constructed the synthetic data set as close as possible to a real scene, real images are often difficult cases because not all of the above assumptions are met. In particular for HBEE-LCNMF, the assumption of every 2 m PAN pixel being pure may not be fulfilled. The next sections will show the results of unmixing for two simulated satellite data sets derived from real acquisitions.

**Table 4.** Mean performance of each unmixing method applied to the synthetic data set. The mean performance values of HBEE-LSNMF have been calculated over the 6 endmembers estimated by the method.

| | HBEE LCNMF | HBEE LSNMF | N-FINDR | ATGP | VCA | SISAL | MVCNMF | JBEELU |
|---|---|---|---|---|---|---|---|---|
| $\overline{NRMSE}_{S,\hat{S}}$ | $\mathbf{3.6{\times}10^{-2}}$ | $7.0{\times}10^{-2}$ | $7.6{\times}10^{-2}$ | $1.4{\times}10^{-1}$ | $1.0{\times}10^{-1}$ | $6.2{\times}10^{-1}$ | $9.8{\times}10^{-2}$ | $2.6{\times}10^{-1}$ |
| $\overline{RMSE}_{S,\hat{S}}$ | $\mathbf{8.9{\times}10^{-3}}$ | $3.4{\times}10^{-2}$ | $2.0{\times}10^{-2}$ | $2.9{\times}10^{-2}$ | $2.7{\times}10^{-2}$ | $8.8{\times}10^{-2}$ | $2.6{\times}10^{-2}$ | $6.6{\times}10^{-2}$ |
| $\overline{SAM}_{S,\hat{S}}(^{o})$ | $\mathbf{1.9{\times}10^{0}}$ | $6.9{\times}10^{0}$ | $4.3{\times}10^{0}$ | $5.2{\times}10^{0}$ | $4.2{\times}10^{0}$ | $2.2{\times}10^{1}$ | $3.9{\times}10^{0}$ | $1.0{\times}10^{1}$ |
| $\overline{SID}_{S,\hat{S}}$ | $\mathbf{2.9{\times}10^{-3}}$ | $8.7{\times}10^{-2}$ | $2.0{\times}10^{-2}$ | $2.7{\times}10^{-2}$ | $2.7{\times}10^{-2}$ | $1.8{\times}10^{-1}$ | $2.6{\times}10^{-2}$ | $1.3{\times}10^{-2}$ |
| $\overline{NRMSE}_{X,\hat{X}}$ | $\mathbf{2.5{\times}10^{-1}}$ | $3.7{\times}10^{-1}$ | $6.3{\times}10^{-1}$ | $9.5{\times}10^{-1}$ | $4.3{\times}10^{-1}$ | $2.2{\times}10^{0}$ | $4.3{\times}10^{-1}$ | $1.7{\times}10^{0}$ |
| $\overline{RMSE}_{X,\hat{X}}$ | $3.0{\times}10^{-2}$ | $\mathbf{2.9{\times}10^{-2}}$ | $3.5{\times}10^{-2}$ | $7.8{\times}10^{-2}$ | $4.2{\times}10^{-2}$ | $3.4{\times}10^{-1}$ | $4.2{\times}10^{-2}$ | $1.8{\times}10^{-1}$ |
| time (s) | $8.9{\times}10^{0}$ | $1.1{\times}10^{0}$ | $1.0{\times}10^{0}$ | $8.8{\times}10^{-1}$ | $9.1{\times}10^{-1}$ | $\mathbf{9.6{\times}10^{-2}}$ | $2.6{\times}10^{0}$ | $4.7{\times}10^{1}$ |

The values in bold are the best for each line.

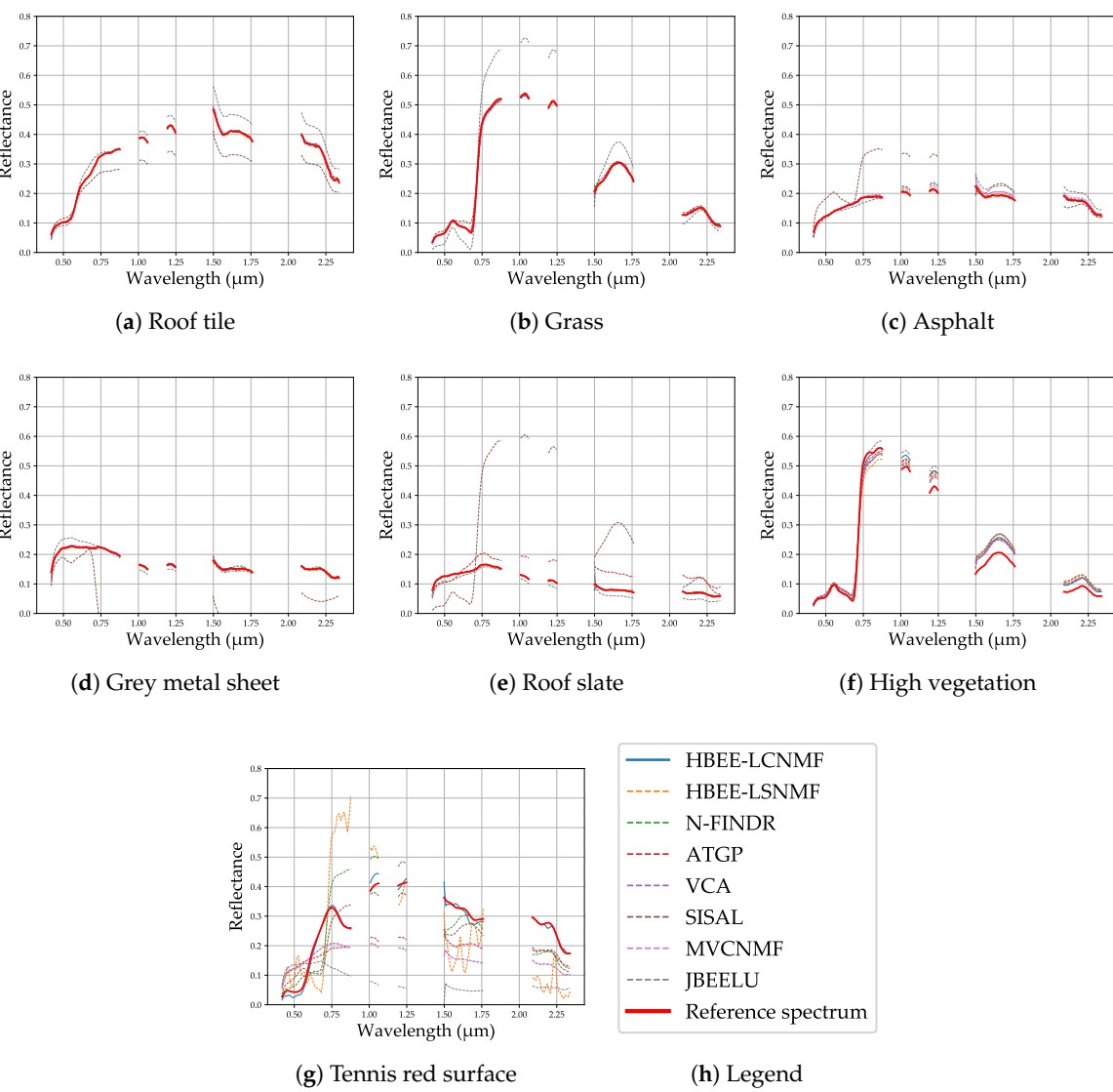

(**a**) Roof tile

(**b**) Grass

(**c**) Asphalt

(**d**) Grey metal sheet

(**e**) Roof slate

(**f**) High vegetation

(**g**) Tennis red surface

(**h**) Legend

**Figure 10.** Estimated endmembers compared to their corresponding references. The (**a**–**e**) plots correspond to materials present in pure pixels and the (**f**,**g**) plots correspond to highly mixed endmembers. The SAM criterion is used to match the estimated endmembers with the reference spectra.

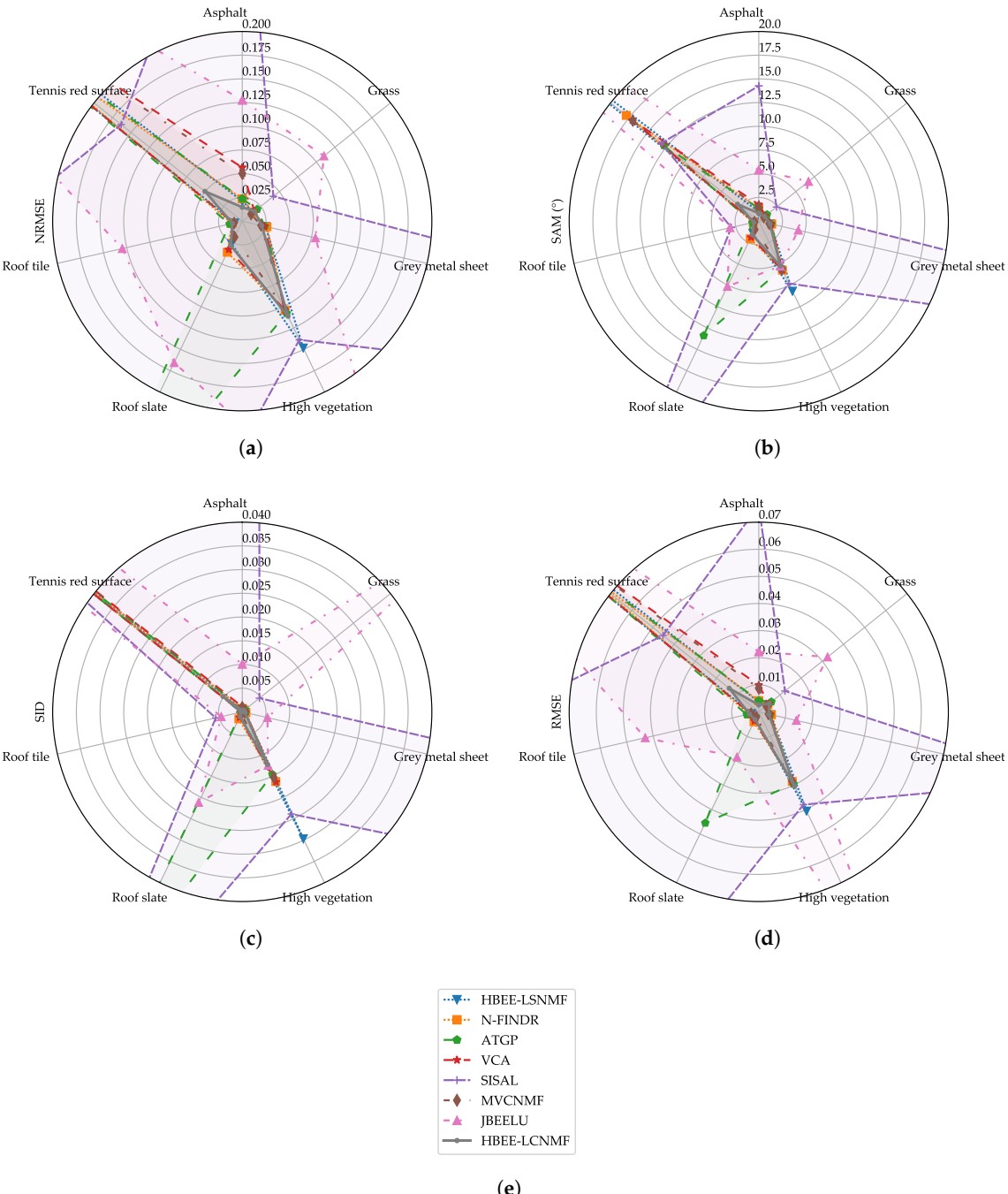

**Figure 11.** Normalized Root Mean Square Error (NRMSE) (**a**), spectral angle measure (SAM) (**b**), Spectral Information Divergence (SID) (**c**), and Root Mean Square Error (RMSE) (**d**) performances for each estimated endmember for the synthetic data set. The legend is given in (**e**).

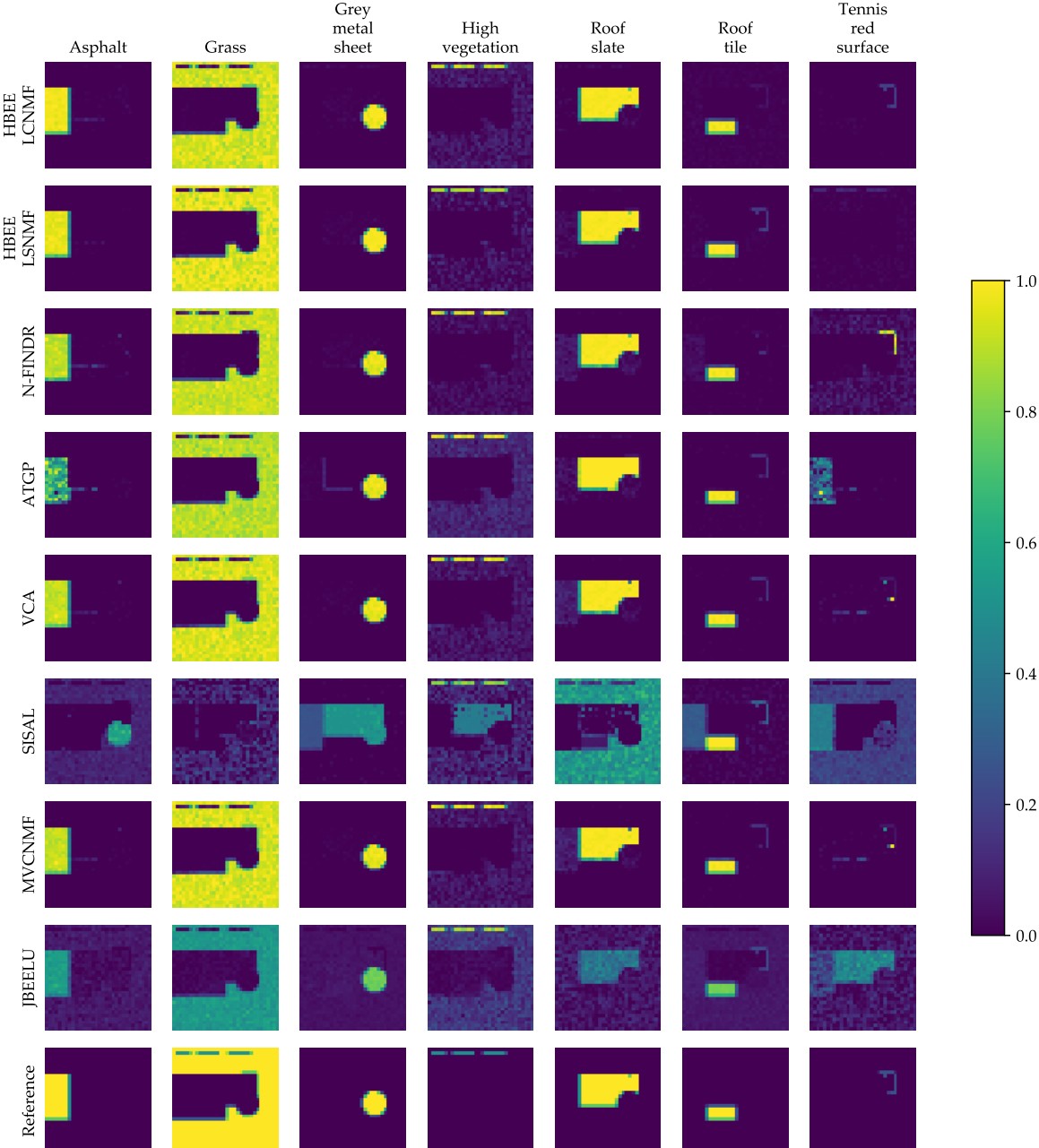

**Figure 12.** Abundance fractions of the seven endmembers (column) estimated with different unmixing methods (row) from the synthetic image, based on the SAM estimate-reference spectrum couples.

## *5.2. Unmixing Results for the Basel Data Set*

Here, HBEE-LCNMF is applied with the parameters given in Table 5 and successfully estimates 6 endmembers.

**Table 5.** HBEE-LCNMF parameter values for the Basel data set.

|  | $\alpha_h$ | $\alpha_d$ | $\alpha_{RE}$ | Stopping NMF Error Value | NMF Max Iteration |
|---|---|---|---|---|---|
| value | 10 (W/m$^2$/sr) | 4.8° | $5.31 \times 10^{-2}$ | $10^{-8}$ | $10^4$ |

The HBEE-LSNMF method has been applied with the parameters given in Table 6 and estimated 5 endmembers.

**Table 6.** HBEE-LSNMF parameter values for the Basel data set.

| | $t_h$ | $t_{sa}$ | $t_{RE}$ | Stopping NMF Error Value | NMF Max Iteration | a |
|---|---|---|---|---|---|---|
| value | 11 W/m$^2$/sr | 10° | 0.05 | $10^{-7}$ | $10^5$ | 6 |

In this image, the green fence and green track materials are green (close spectra in the visible range) and are petroleum-based synthetic materials, leading to quite similar spectra in SWIR. Similarly to HBEE-LCNMF, the HYSIME method estimates that there are 6 endmembers, whereas HFC estimates a number of endmembers equal to 5, then 5 and 4 for false alarm thresholds being, respectively, $10^{-3}$, then $10^{-4}$ and $10^{-5}$. Due to the similarity between the spectra of the two green synthetic materials and since they are highly mixed in the 8 m GSD image, it is reasonable to look for a single endmember for both and to set the number of endmembers to 6 instead of 7. This is also consistent with the number of endmembers estimated by HYSIME, HFC and HBEE-LCNMF.

The estimated endmembers are given in Figure 13.

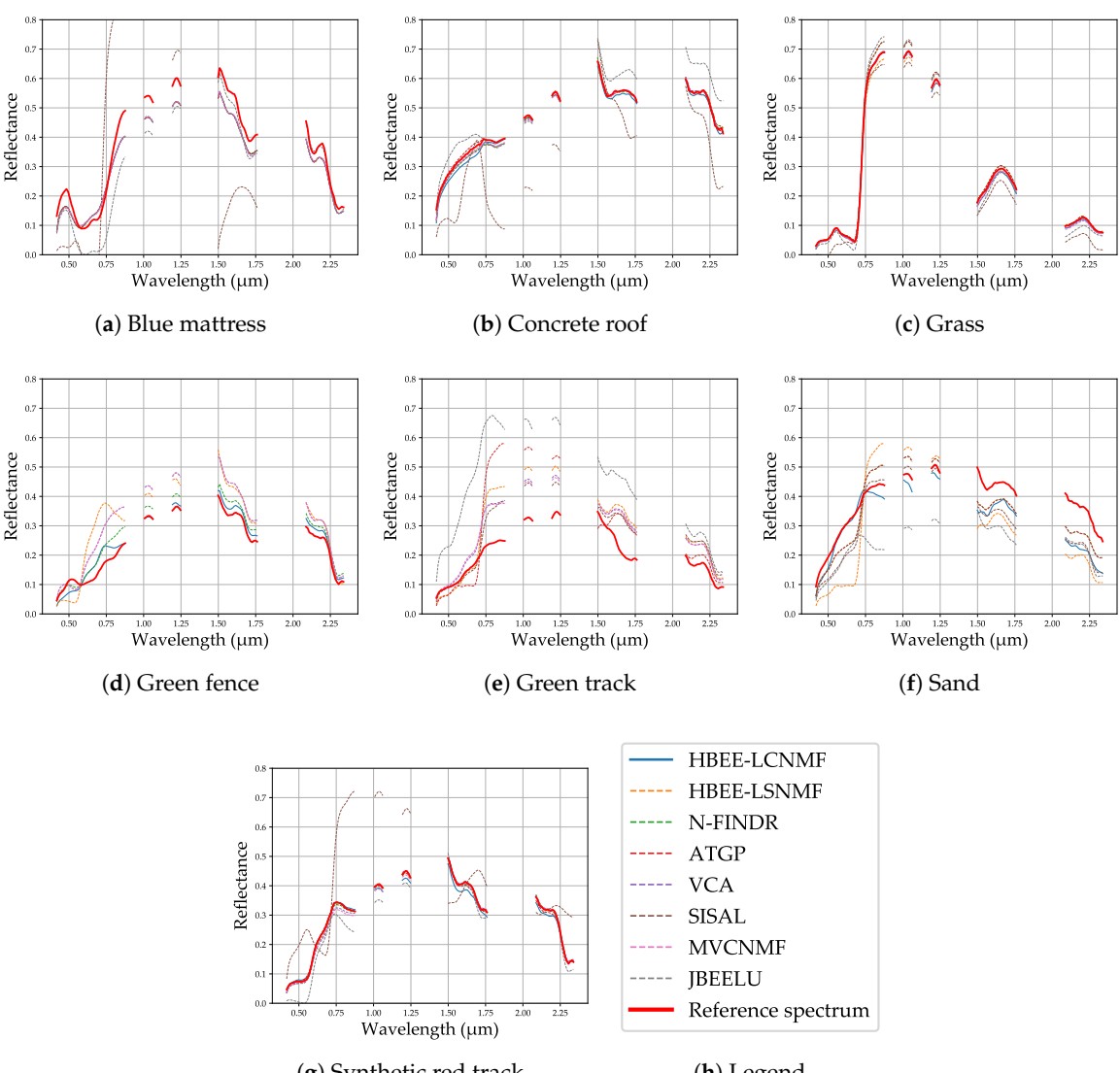

**Figure 13.** Estimated endmembers compared to their corresponding references for the Basel data set. The SAM criterion is used to match the estimated endmembers with the reference spectra.

First, the heterogeneity assumption used for HBEE has been assessed over several scenes (urban and rural). The first 5% most homogeneous HS pixels are always pure, validating our hypothesis.

Yet, false positives (detection of non-pure pixels as pure ones) can occur for the following reasons: first, if two materials have very similar panchromatic radiance levels, the PAN pixels where they are located may have equivalent values. Consequently, their heterogeneity criterion value may be low. Nevertheless, these heterogeneity values are not the lowest ones and the selection of the corresponding spectra as endmembers is unlikely. Secondly, false positives may occur if some PAN pixels are non-pure. This problem could be overcomed if the PAN image would have a better spatial resolution, which would increase the number of PAN pixels per HS pixel, and would reduce the mixed-pixel to pure-pixel ratio in the PAN image. The HBEE stage extracts spectra corresponding to the blue mattress, concrete roof, grass, and synthetic red track while the LCNMF stage estimates the sand and green fence materials. The LCNMF stage does not estimate the green track material. As a matter of fact, the green track, the synthetic red track and the blue mattress material share the same spectral characteristics in the SWIR. Since the green track is always highly mixed with the synthetic red track material, and the spectrum of the latter and the blue mattress are already extracted by HBEE, the reconstruction error value remains low and the corresponding area is therefore not detected by LCNMF. This phenomenon can be seen in Figure 14 as a small proportion of the blue mattress material is detected by HBEE-LCNMF in the location where the green track and synthetic red track materials are present.

The performance for each estimate-reference spectrum couple is shown in Figure 15. The estimate-reference couples may differ between criteria, as the matching procedure is performed for each criterion independently.

The pure-pixel based state-of-the-art methods extract slightly better spectra than those of HBEE. This is likely due to the noise present in the PAN image which influences the heterogeneity criterion values $\eta$, especially for the lowest ones. Nevertheless, the endmembers estimated with HBEE are close to the reference spectra in terms of shape and level. The LCNMF stage correctly estimates a spectrum corresponding to the synthetic red track material and outperforms the other methods for the estimation of the sand spectrum. The abundance fractions are provided in Figure 14. Those estimated by HBEE-LCNMF are well estimated and their locations are consistent with the ground truth.

The mean performances are shown in Table 7. For this data set, HBEE-LCNMF, N-FINDR, MVCNMF, and VCA exhibit equivalent performances, whereas the other methods are most often much less accurate.

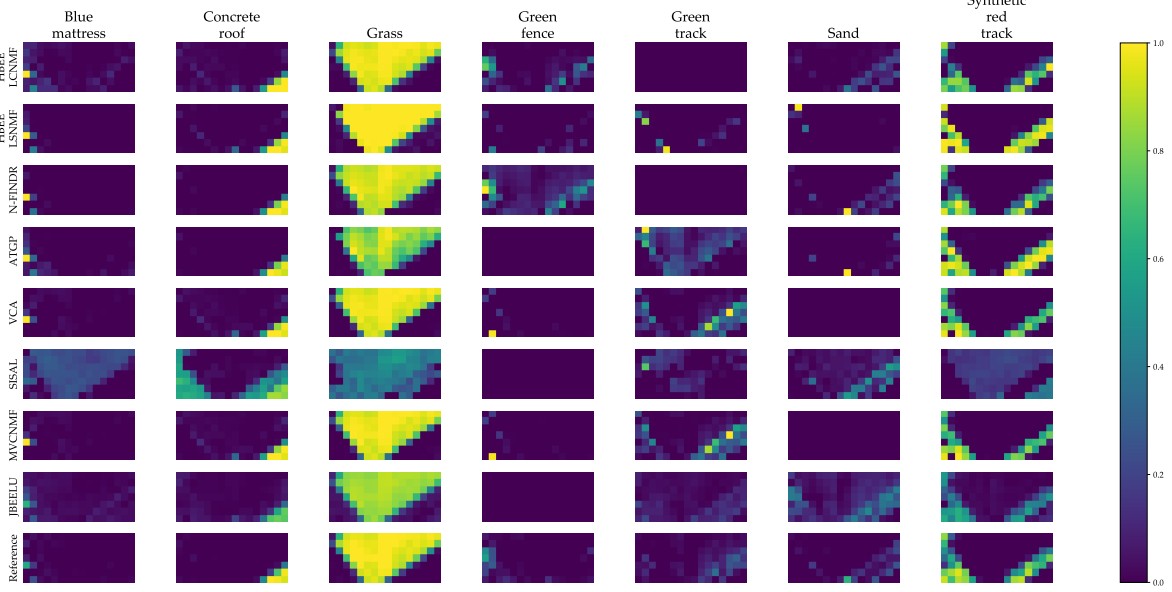

**Figure 14.** Estimated abundance fractions for the Basel data set based on the SAM of the estimate-reference spectrum couples.

**Table 7.** Compared mean performances of the unmixing methods for the Basel data set. The mean has been computed with the 6 endmembers estimated by the methods.

| | HBEE LCNMF | HBEE LSNMF | N-FINDR | ATGP | VCA | SISAL | MVCNMF | JBEELU |
|---|---|---|---|---|---|---|---|---|
| $\overline{NRMSE}_{S,\hat{S}}$ | $\mathbf{9.3 \times 10^{-2}}$ | $2.2 \times 10^{-1}$ | $\mathbf{9.3 \times 10^{-2}}$ | $1.7 \times 10^{-1}$ | $1.3 \times 10^{-1}$ | $4.3 \times 10^{-1}$ | $1.3 \times 10^{-1}$ | $1.9 \times 10^{-1}$ |
| $\overline{RMSE}_{S,\hat{S}}$ | $3.3 \times 10^{-2}$ | $5.9 \times 10^{-2}$ | $\mathbf{3.2 \times 10^{-2}}$ | $4.8 \times 10^{-2}$ | $4.2 \times 10^{-2}$ | $1.3 \times 10^{-1}$ | $4.2 \times 10^{-2}$ | $6.6 \times 10^{-2}$ |
| $\overline{SAM}_{S,\hat{S}}(^o)$ | $3.6 \times 10^{0}$ | $7.2 \times 10^{0}$ | $\mathbf{3.3 \times 10^{0}}$ | $5.6 \times 10^{0}$ | $3.5 \times 10^{0}$ | $1.8 \times 10^{1}$ | $3.4 \times 10^{0}$ | $7.3 \times 10^{0}$ |
| $\overline{SID}_{S,\hat{S}}$ | $9.9 \times 10^{-3}$ | $4.9 \times 10^{-2}$ | $\mathbf{7.6 \times 10^{-3}}$ | $2.8 \times 10^{-2}$ | $8.7 \times 10^{-3}$ | $8.0 \times 10^{-2}$ | $8.3 \times 10^{-2}$ | $5.6 \times 10^{-2}$ |
| $\overline{NRMSE}_{X,\hat{X}}$ | $\mathbf{4.8 \times 10^{-1}}$ | $8.4 \times 10^{-1}$ | $5.1 \times 10^{-1}$ | $7.1 \times 10^{-1}$ | $7.4 \times 10^{-1}$ | $1.5 \times 10^{0}$ | $7.4 \times 10^{-1}$ | $6.4 \times 10^{-1}$ |
| $\overline{RMSE}_{X,\hat{X}}$ | $\mathbf{5.5 \times 10^{-2}}$ | $1.1 \times 10^{-1}$ | $6.2 \times 10^{-2}$ | $1.0 \times 10^{-1}$ | $8.7 \times 10^{21}$ | $2.6 \times 10^{-1}$ | $8.7 \times 10^{-2}$ | $9.0 \times 10^{-2}$ |
| time (s) | $2.7 \times 10^{0}$ | $1.5 \times 10^{-1}$ | $2.1 \times 10^{-1}$ | $1.2 \times 10^{-1}$ | $1.3 \times 10^{-1}$ | $5.8 \times 10^{-2}$ | $1.4 \times 10^{0}$ | $1.3 \times 10^{-1}$ |

The values in bold are the best for each line.

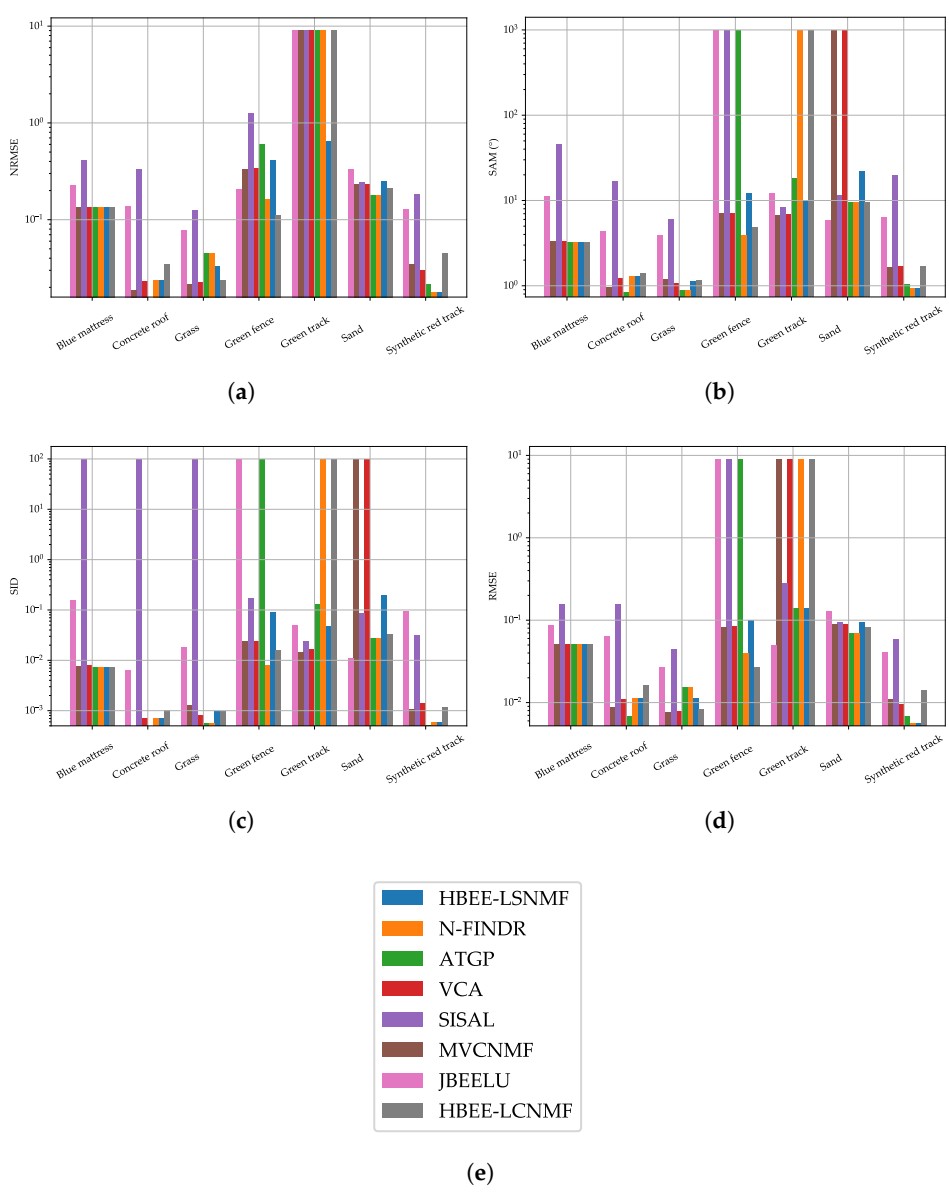

(a)　　　　　　　　　　　　　(b)

(c)　　　　　　　　　　　　　(d)

(e)

**Figure 15.** NRMSE (**a**), SAM (**b**), SID (**c**), and RMSE (**d**) performances for each estimated endmember for the Basel data set. Non-estimated endmembers are shown with values set to 9 for NRMSE, 9 for RMSE, 999 for SAM, and 99 for SID. Invisible bars mean the criterion values are very low. Legend is given in (**e**).

## 5.3. Unmixing Results for the Mauzac Data Set

The HBEE-LCNMF method has been applied with the parameter values provided in Table 8. These values are quite close to those used for the Basel data set.

**Table 8.** HBEE-LCNMF parameter values for the Mauzac data set.

|       | $\alpha_h$        | $\alpha_d$ | $\alpha_{RE}$      | Stopping NMF Error Value | NMF Max Iteration |
|-------|-------------------|------------|--------------------|--------------------------|-------------------|
| value | 8 (W/m$^2$/sr)    | 6°         | $6 \times 10^{-2}$ | $10^{-7}$                | $10^5$            |

The HBEE-LSNMF method has been applied with the parameter values provided in Table 9 and estimates 4 endmembers.

**Table 9.** HBEE-LSNMF parameter values for the Mauzac data set.

|       | $t_h$          | $t_{sa}$ | $t_{RE}$ | Stopping NMF Error Value | NMF Max Iteration | a |
|-------|----------------|----------|----------|--------------------------|-------------------|---|
| value | 12 W/m$^2$/sr  | 10°      | 0.1      | $10^{-7}$                | $10^5$            | 4 |

For this data set, HYSIME estimates a number of pure materials equal to 12 and HFC estimates this number to 6, 5 and 5 for false alarm thresholds being, respectively, $10^{-3}$, $10^{-4}$, and $10^{-5}$. But, as mentioned in Section 3.2, only 5 distinct materials are present in the scene. HBEE-LCNMF estimates all 5 endmembers. The estimated endmembers are shown in Figure 16.

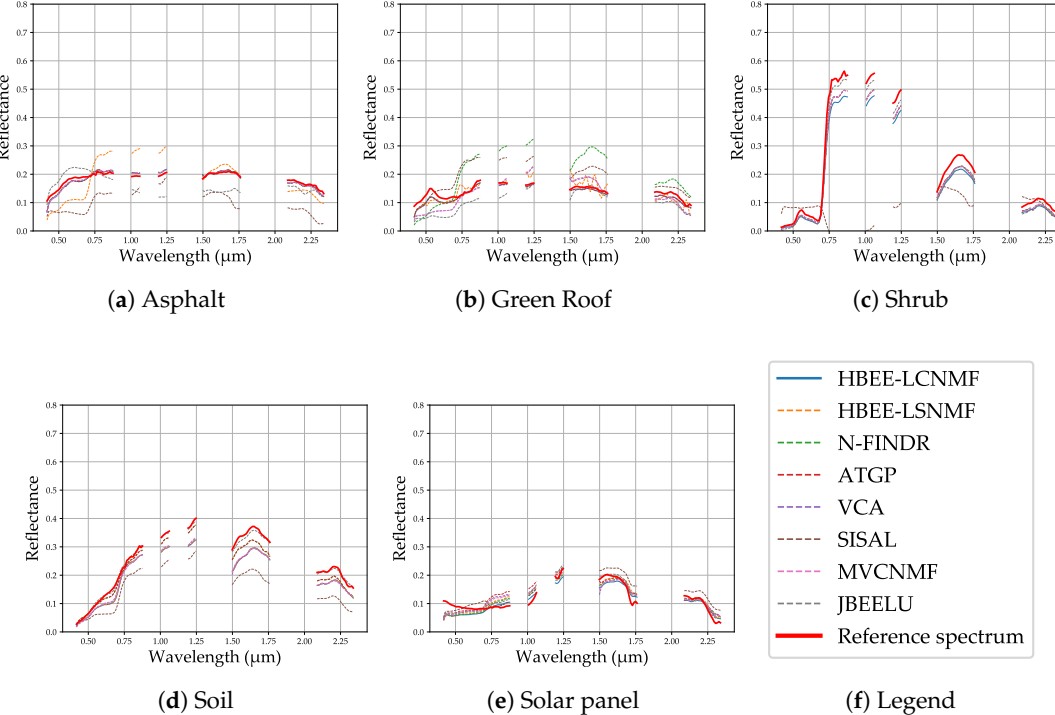

(**a**) Asphalt     (**b**) Green Roof     (**c**) Shrub

(**d**) Soil     (**e**) Solar panel     (**f**) Legend

**Figure 16.** Estimated endmembers and their corresponding references for the Mauzac data set. The SAM criterion is used to match the estimated endmembers with the reference spectra.

The HBEE stage extracts only two endmembers, which is a difficult case for the LCNMF stage. Nevertheless, all the remaining endmembers are successfully retrieved by LCNMF. Moreover, the green roof material is located inside only one 8 m GSD HS pixel. This pixel is a mixed one since it also contains soil material and shadow. However, due to the weak spectral contribution of shadow and the relative small abundance of the soil material, geometrical methods can find also an endmember

close to the reference spectrum, as it is the case for ATGP. But the presence of shadow, which is not accounted for in the LMM, induces a high contrast in the PAN image; thus, this pixel is not identified as a pure one by HBEE.

Regarding the solar panels, they are only present in mixed pixels where their shadows are projected over the soil. Then, pure-pixel based unmixing methods fail to extract this endmember spectrum due to the lack of related pure pixels, and the other linear unmixing methods also fail to estimate it because the shadows are not accounted for in the LMM.

Nevertheless, the LCNMF stage delivers a better estimate of the solar panel spectrum than the concurrent unmixing methods, except for JBEELU, in terms of performance, as shown in Figure 17.

HBEE-LCNMF estimates the other endmembers with performance equivalent to those of the other methods.

The abundance fractions, provided in Figure 18, related to the HBEE-LCNMF method are well estimated in terms of location of the materials.

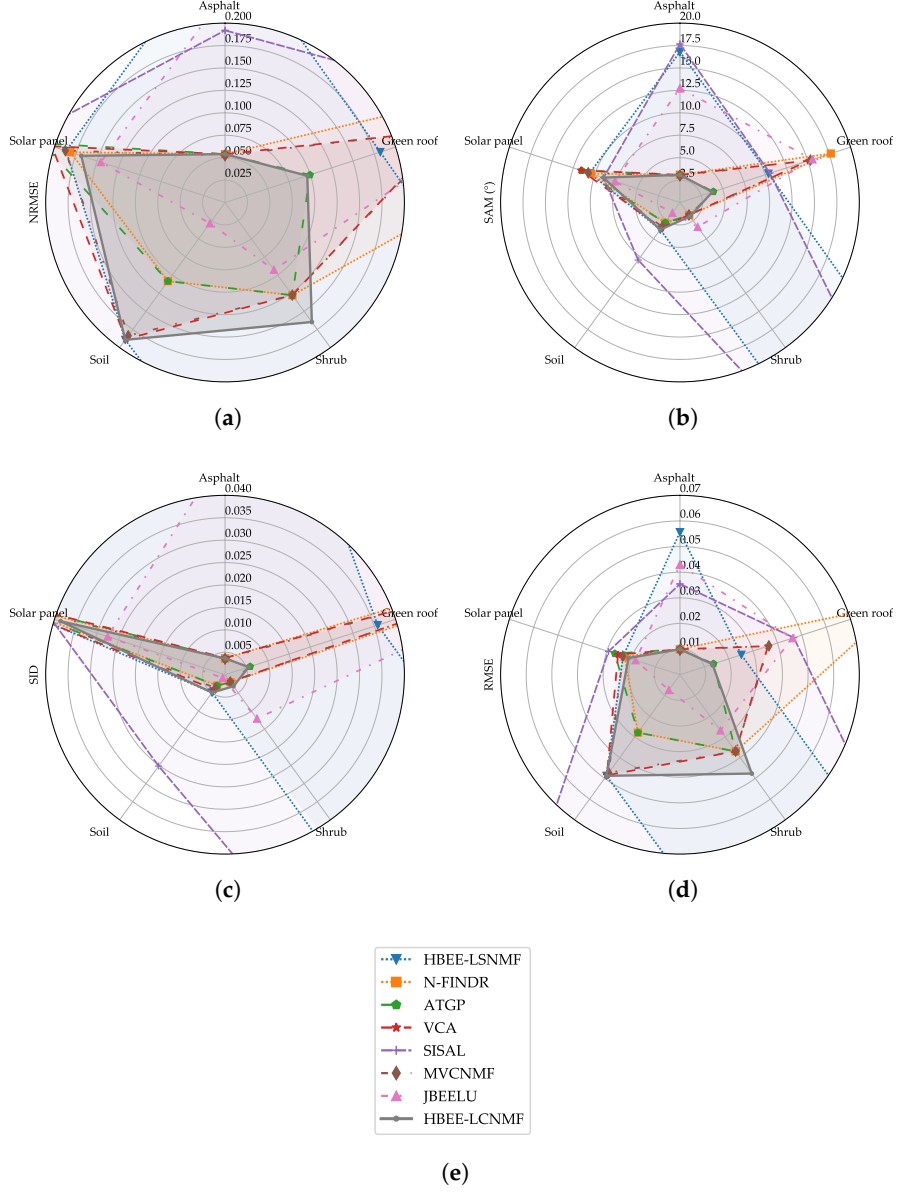

**Figure 17.** NRMSE (**a**) , SAM (**b**), SID (**c**), and RMSE (**d**) performances for each estimated endmember for the Mauzac data set. Legend is given in (**e**).

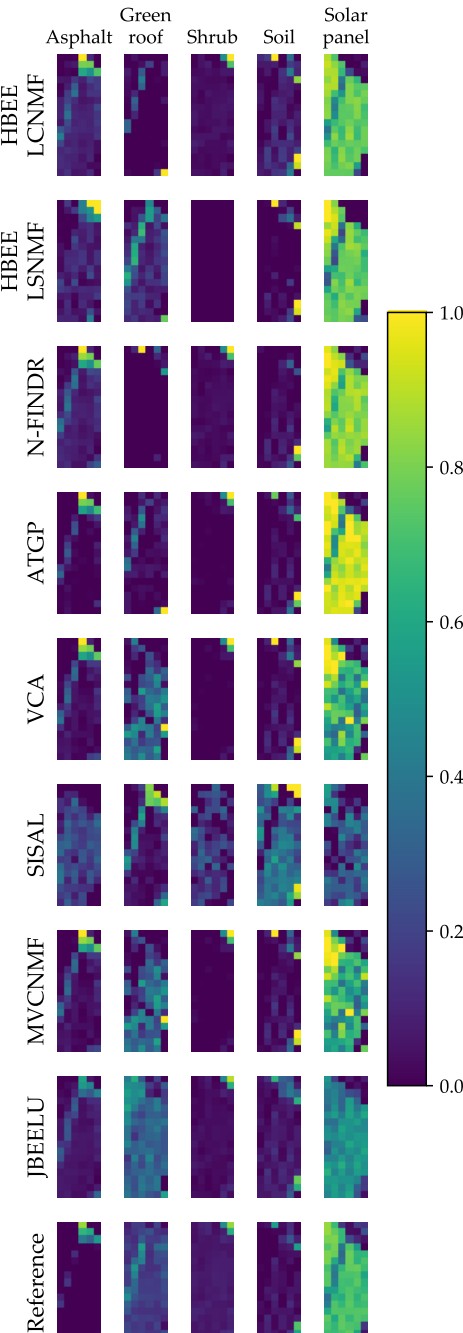

**Figure 18.** Estimated abundance fractions for the Mauzac data set based on the SAM estimate-reference spectrum couples.

The mean performances are detailed in Table 10. That of HBEE-LCNMF is equivalent to that of ATGP and better than those of the other methods. Moreover, HBEE-LCNMF achieves a satisfactory spectral unmixing without knowing the number of endmembers in advance.

**Table 10.** Compared mean performances for each method applied to the Mauzac data set. The mean performance of HBEE-LSNMF has been calculated with the 4 endmembers estimated by this method.

| | HBEE LCNMF | HBEE LSNMF | N-FINDR | ATGP | VCA | SISAL | MVCNMF | JBEELU |
|---|---|---|---|---|---|---|---|---|
| $\overline{NRMSE}_{S,\hat{S}}$ | $1.3 \times 10^{-1}$ | $2.1 \times 10^{-1}$ | $2.2 \times 10^{-1}$ | $\mathbf{1.2} \times 10^{-1}$ | $1.7 \times 10^{-1}$ | $4.1 \times 10^{-1}$ | $1.6 \times 10^{-1}$ | $1.7 \times 10^{-1}$ |
| $\overline{RMSE}_{S,\hat{S}}$ | $2.8 \times 10^{-2}$ | $3.8 \times 10^{-2}$ | $3.7 \times 10^{-2}$ | $\mathbf{2.3} \times 10^{-2}$ | $3.1 \times 10^{-2}$ | $9.4 \times 10^{-2}$ | $3.1 \times 10^{-2}$ | $2.8 \times 10^{-2}$ |
| $\overline{SAM}_{S,\hat{S}}(^{o})$ | $\mathbf{4.3} \times 10^{0}$ | $1.0 \times 10^{1}$ | $7.1 \times 10^{0}$ | $4.6 \times 10^{0}$ | $7.0 \times 10^{0}$ | $2.2 \times 10^{1}$ | $6.8 \times 10^{0}$ | $8.1 \times 10^{0}$ |
| $\overline{SID}_{S,\hat{S}}$ | $\mathbf{1.1} \times 10^{-2}$ | $5.0 \times 10^{-2}$ | $4.4 \times 10^{-2}$ | $1.4 \times 10^{-2}$ | $3.1 \times 10^{-2}$ | $1.1 \times 10^{-1}$ | $2.9 \times 10^{-2}$ | $3.7 \times 10^{-2}$ |
| $\overline{NRMSE}_{X,\hat{X}}$ | $5.4 \times 10^{-1}$ | $6.6 \times 10^{-1}$ | $6.1 \times 10^{-1}$ | $\mathbf{4.4} \times 10^{-1}$ | $6.3 \times 10^{-1}$ | $1.4 \times 10^{0}$ | $6.3 \times 10^{-1}$ | $5.0 \times 10^{-1}$ |
| $\overline{RMSE}_{X,\hat{X}}$ | $\mathbf{1.0} \times 10^{-1}$ | $1.2 \times 10^{-1}$ | $1.3 \times 10^{-1}$ | $1.1 \times 10^{-1}$ | $1.4 \times 10^{-1}$ | $2.8 \times 10^{-1}$ | $1.4 \times 10^{-1}$ | $1.2 \times 10^{-1}$ |
| time (s) | $1.1 \times 10^{0}$ | $9.9 \times 10^{-1}$ | $2.1 \times 10^{-1}$ | $9.2 \times 10^{-2}$ | $1.1 \times 10^{-1}$ | $5.9 \times 10^{-2}$ | $7.6 \times 10^{-1}$ | $1.3 \times 10^{1}$ |

The values in bold are the best for each line.

## 6. Conclusions

In this study, a new unmixing method using both a hyperspectral image and a co-registered PAN image has been presented. It is composed of two stages called HBEE and LCNMF. The HBEE stage extracts a first set of endmembers which are represented by pure pixels by means of a heterogeneity criterion applied to the panchromatic image and followed by a hierarchical spectral clustering. The LCNMF stage estimates the remaining endmembers that are not represented by pure pixels. It consists in applying a sequence of constrained NMFs to the areas containing the remaining endmembers. These areas are selected from the reconstruction error maps that are computed with the endmembers and the abundance fractions estimated during the unmixing processes. Areas with high values of reconstruction error are assumed to contain endmembers that were not yet extracted.

The proposed method was first tested on a synthetic data set generated with real material spectra, taking into account the spectral variability that can be observed for a given material, and a realistic noise model. The HS image was designed to contain both pure-pixel-represented endmembers, as well as endmembers that are not. This synthetic data set was generated with the instrument characteristics of the HYPXIM/HYPEX-2 space mission. This first test has shown the efficiency of the method, especially in estimating the highly mixed endmembers, and its reliability in retrieving the correct number of endmembers in the image. For this data set, this new method outperforms the tested state-of-the art methods.

A second evaluation was conducted on two scenes on which a ground truth can be built: the number of endmembers is known, as well as their spectral reflectances, and their abundances in each pixel can be roughly estimated using FCLS. These scenes have been acquired by airborne sensors and the corresponding HS images have been simulated to the HYPXIM/HYPEX-2 features, resulting in a 2 m GSD PAN image and an 8 m GSD HS image for each scene. The HBEE-LCNMF method yields promising results with performance equivalent to or better than the state-of-the-art methods, without knowing the number of endmembers in advance. For one scenario, the proposed approach successfully retrieves 6 endmembers with a similar performance compared to N-FINDR, MVCNMF, and VCA. For the other scenario, all the endmembers are well estimated and HBEE-LCNMF has the best performances with ATGP. Moreover, the tested state-of-the-art methods have been applied with the correct number of endmembers to be estimated while the HYSIME method largely over-estimates this number on the second real data set. In the end, the proposed method always shows robust performance for each scenario without knowing the number of endmembers to be estimated, while the tested state-of-the-art methods have fluctuating performance.

On the basis of the tests carried out so far and the identified limitations, several perspectives have emerged to extend this work. Although this study focused on the HYPXIM-HYPEX-2 instrument specifications, the use of PAN images with a GSD smaller than 2 m may enhance the detection of pure pixels and the selection of endmembers from the associated HS pixels. Moreover, some materials may have the same panchromatic luminance value. To tackle this problem, the use of a multispectral image instead of the PAN image may be relevant.

**Author Contributions:** Methodology, S.R., V.A., X.B., Y.D. and S.M.; software, S.R.; validation, S.R., V.A., X.B., Y.D. and S.M.; data curation, S.R.; writing–original draft preparation, S.R.; writing–review and editing, S.R., V.A., X.B., Y.D. and S.M. All authors have read and agreed to the published version of the manuscript.

**Funding:** This work is co-funded by ONERA, the French aerospace lab and the Centre National d'Études Spatiales (CNES). This research received no external funding.

**Acknowledgments:** We thank the CNES and more precisely Vincent Lonjou for providing us with the HYPXIM/HYPEX-2 sensor model features. We also deeply thank Christian Feigenwinter, Andreas Hueni and Michael E. Schaepman for providing us with the Basel database and answering our questions. We thank Philippe Deliot from ONERA for providing us with the Mauzac HS acquisition.

**Conflicts of Interest:** The authors declare no conflict of interest.

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
