# Peer review of "Using a Panchromatic Image to Improve Hyperspectral Unmixing"

_remotesensing, doi:10.3390/rs12172834_

Round 1

Reviewer 1 Report

This paper presents a new hyperspectral unmixing method by the assistance of a panchromatic image. Firstly, a set of endmembers is estimated based on a heterogeneity criterion applied on the panchromatic image followed by a spectral clustering. And then, a local approach using a constrained non-negative matrix factorization strategy is proposed to complete this first endmember set. Its performance is compared to those of state-of-the-art methods obtained on synthetic and satellite data. Overall, this is well-written paper and a good job. I have some minor issues:

There is only one band for the panchromatic image, how to guarantee the accuracy of the obtained endmembers. I suggest the author give more analysis.

To demonstrate the effectiveness of the proposed improvements, ablation studies are needed. For example, how does the introduced panchromatic image effect the performance? Why do you choose the ascending hierarchical clustering technique?

Since [52] is the previous of the authors, it should be compared in the experiments.

Reviewer 2 Report

The article entitled “Using a Panchromatic Image to Improve Hyperspectral Unmixing” presents a new approach, called Heterogeneity Based Endmember Extraction (HBEE) - Local Constrained Non-negative Matrix Factorisation (LCNMF). In this paper, the authors introduce the use of a co-registered panchromatic image in the unmixing process that thanks to its higher GSD improve the selection of the endmember spectra contained in the HS imagery. The proposed method has been implemented by the authors with the aim to overcome the need for prior knowledge of at least one pure pixel for each material and applied to two datasets surveyed using an aerial platform simulating the forthcoming hyperspectral satellite HYPXIM/HYPEX-2 data.

I read with interest the proposed research that deals with an issue worthwhile to investigate. Moreover, I have to say that in reviewing this paper, what I really appreciated is that the authors provided a strong methodological section. Furthermore, they provided additional information handy for those researchers who are not familiar with hyperspectral imagery and the issue of spectral mixing and unmixing. As a reviewer, I appreciate it when a research article provides a comprehensive picture of the covered issue.

General comments

The present manuscript is well organized and has a good style. Also, the English style is fine. Having said that, I definitely agree with the acceptance of the present paper after providing the corrections provided in the following rows and that I consider as minor.

ABSTRACT

It not advisable to introduce acronyms without any explanation (e.g., HBEE-LCNMF, VCA, N-FINDR). Moreover, I suggest adding in what case study the proposed method has been applied.

KEYWORDS

Maybe few and too general. Keywords and title should help a researcher to find the paper of his/her own interest.

  1. INTRODUCTION

Well-written, appropriate references cited and aims of the proposed research clearly stated.

  1. Proposed method

Well-written and reporting the necessary information to understand the proposed method. By the way, I would suggest the authors provide some information about the used hyperspectral data (i.e., the forthcoming data characteristic of the French mission HYPXIM/HYPEX-2). I think this information could be useful for a potential reader.

  1. Materials

Figure 7 and figure 8. To improve the comparison of the provided graphs, the y-axes should have the same scale.

  1. Performance assessment protocol

Good in its present form.

  1. Results and discussion

Figure 10 and figure 16. To improve the comparison of the provided graphs, the y-axes should have the same scale.

  1. Conclusion

Good in its present form.

Reviewer 3 Report

The paper proposes a two-step unmixing process which estimates near pure pixels as its first step. Clustering as a pre-processing step is performed in KMSCD unmixing algorithm (DOI: 10.3390/jimaging5110085) and an argument can be made for the added benefit of PAN image input.

In my opinion, the overall presentation of methods and results can be improved. The proposed method compares with methods which (most if not all) were published two decades ago, and the paper needs more relevant publications. Example include the minimum volume algorithm (DOI: 10.1109/TGRS.2017.2728104) can be used instead of/in addition to MVCNMF algorithm published 2007. Additionally, the following are my recommendations/correction:

  • Tables 3 and 5, for instance, shows only normalised results and a metric needs to be added for absolute value (such as RMSE)
  • Please include a figure/table to show the robustness of the proposed algorithm to increase in added noise
  • Presentation of the results can be optimised

Lastly, public availability of the source code of the proposed method is always welcome should the authors/institute wish it.

Round 2

Reviewer 3 Report

The authors provided satisfactory responses to my comments. I recommend adding appropriate explanations provided to the manuscript for issues #1 (preferably in the introduction for a thorough background and adding unmixing using a sliding window for local) and #2 (because (i) the reason why VCA and PPI algorithms perform poorly in the datasets mentioned is intuitive from the abundance figures, and (ii) even with recent methods published with similar or with improved accuracy for methods like the minimum volume and Bayesian, it is surprising why former publications are given preference in section 4.2). Aside from that, I recommend acceptance of the manuscript. Good luck.
